# CAN EXTERNAL VALIDATION TOOLS IMPROVE ANNOTATION QUALITY FOR LLM-AS-A-JUDGE?

## ABSTRACT

Pairwise preferences over model responses are widely collected to evaluate and provide feedback to *large language models* (LLMs). Given two alternative model responses to the same input, a human or AI annotator selects the *"better"* response. This approach can provide feedback for domains where other hard-coded metrics are difficult to obtain (e.g., quality of a chat interactions), thereby helping measure model progress or model fine-tuning (e.g., via *reinforcement learning from human feedback*, RLHF). However, for some domains it can be tricky to obtain such pairwise comparisons in high quality - from AI *and* humans. For example, for responses with many factual statements or complex code, annotators may overly focus on simpler features such as *writing quality* rather the underlying facts or technical details. In this work, we explore augmenting standard AI annotator systems with additional tools to improve performance on three challenging response domains: *long-form factual*, *math* and *code* tasks. We propose a *tool-using agentic system* to provide higher quality feedback on these domains. Our system uses web-search and code execution to ground itself based on *external validation*, independent of the LLM's internal knowledge and biases. We provide extensive experimental results evaluating our method across the three targeted response domains as well as general annotation tasks, using *RewardBench* data (incl. *AlpacaEval* and *LLMBar*), as well as three new datasets for areas where pre-existing datasets are saturated. Our results indicate that external tools can indeed improve AI annotator performance in many, but not all, cases. More generally, our experiments highlight the high variability of AI annotator performance with respect to simple parameters (e.g., prompt) and the need for improved (non-saturated) annotator benchmarks. We share our data and code publicly.[1]

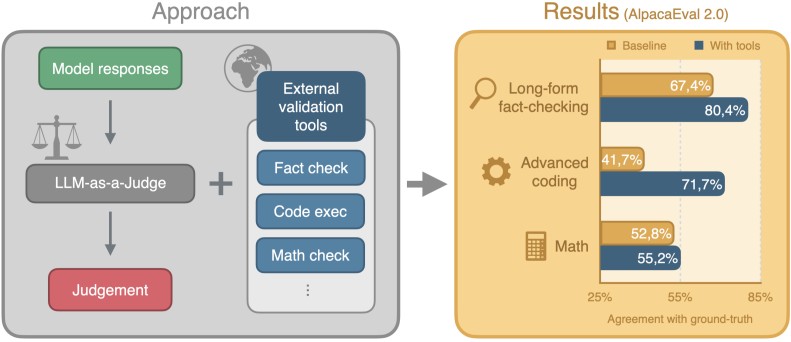

Figure 1: **Summary of our approach and results: we extend standard LLM-as-a-Judge baselines with external validation tools based on web-search and code execution.** We observe that the resulting system is often, but not always, able to improve performance (measured as agreement with ground-truth annotation) across a range of response domains that are typically challenging for LLM-as-a-Judge systems: (1) *long-form factual*, (2) *advanced coding*, and (3) *math* responses. The results shown are based on the popular AlpacaEval 2.0 baseline annotator, full results in Section 4.

---

[1]Link to repository will be shared upon publication.

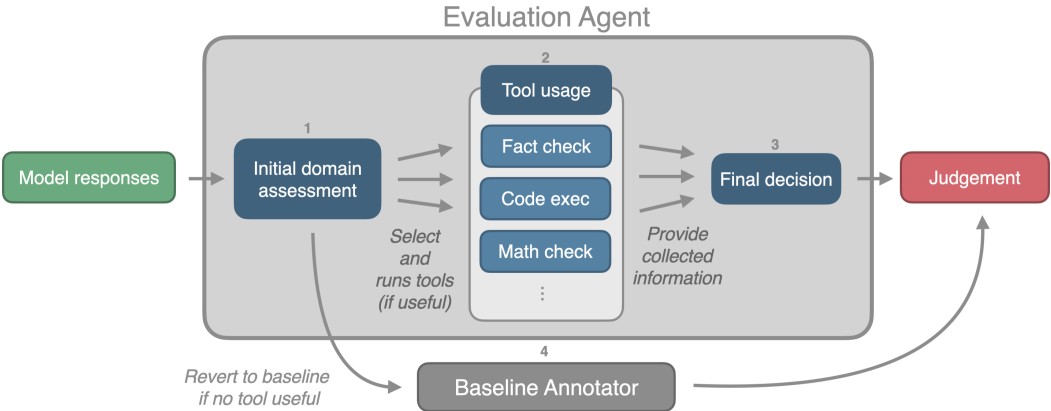

Figure 2: **Overview of our tool-using AI annotator architecture, referred to as *Evaluation Agent*.** In the (1) *initial domain assessment* the appropriate tools are selected for each response (e.g. for a wiki-style text the fact check tool); then, in (2) *tool usage*, each selected tool is run and the tool outputs are combined into a single prompt to make a (3) *final decision*. If none of the tools are selected (i.e., no tool deemed useful), the agent instead reverts and returns an annotation from the (4) *baseline annotator* (e.g., AlpacaEval 2.0).

# 1 INTRODUCTION

Pairwise feedback is widely used to understand LLM performance on complex tasks that more traditional benchmarks fail to measure well. Given a prompt and two possible responses, the annotator decides which response is *"better"*. This pairwise judgement can be used for evaluation (e.g., *Chatbot Arena* (Chiang et al., 2024)) or to provide feedback for training (e.g., via RLHF (Stiennon et al., 2020; Ouyang et al., 2022) or DPO (Rafailov et al., 2023)). Both human and AI annotators (also referred to as *LLM-as-a-Judge*) are used to collect such feedback. Human annotators are often considered higher quality but more expensive.

*Both human and AI annotations have notable limitations:* AI annotators have been observed to be susceptible to a number of biases, including changing preference based on superficial features like *response order* Zheng et al. (2023) or *response length* Dubois et al. (2024)). Whilst possibly providing higher quality annotations than AI annotators, human annotators also have known issues. For example, human annotators have been observed to let their assessment of truthfulness be affected by the assertiveness of responses (Hosking et al., 2024).

In certain domains it is *particularly challenging* to obtain high-quality annotations: for responses containing *long-form factual*, *advanced coding* and *math* content both AI and (many) human annotators struggle to provide reliable annotations (Zheng et al., 2023). Annotating responses in these domains requires expertise and careful deliberation, challenging to achieve for human annotators in a limited amount of time. AI annotators may be less "time-constrained" but nevertheless due to known reliability issues (e.g, hallucinations, limited basic arithmetic) often fail to provide high quality annotations in these domains (Yang et al., 2023).

In this work, we aim to explore improving the annotation quality of widely used AI annotators on these challenging domains by augmenting the annotators with tools that can *externally validate answers*. We enable responses to be fact-checked using *web-search*, or verified using *code execution*. Our setup is illustrated in Figure 2. In particular, we make the following contributions:

1. **Extensible framework for using tools with existing AI annotators**. We introduce a new framework that enables the integration of new tools on top of existing AI annotators to improve annotation quality in certain domains using external validation. Our framework is *agentic* in the sense that an LLM assesses the response domain and plans the optimal tool usage accordingly. We provide a number of initial tool implementations: (1) a *long-form fact checking* tool based on the *Search Augmented Fact Evaluation* (SAFE) method by Wei et al. (2024); (2) a *code check*

tool built on OpenAI's code interpreter API; and (3) a *math check* tool similarly built on code execution. We open-source the corresponding code[2]

2. **New datasets for challenging pairwise annotation tasks.** We share four new pairwise datasets extending domains that are currently saturated or not covered well in existing pairwise annotation benchmarks (such as RewardBench (Lambert et al., 2024)) with more challenging tasks. In particular, we adapt subsets of the *LongFact* (Wei et al., 2024), *TruthfulQA* (Lin et al., 2022), *GSM8k* Cobbe et al. (2021a) and *APPS* (Hendrycks et al., 2021) datasets to the pairwise setting.

3. **Extensive experimental results evaluating our framework's capabilities.** We evaluate our framework's effectiveness across a wide range of tasks including the newly created datasets as well as well-established benchmarks. We compare our method to a number of popular state-of-the-art AI annotators, including the annotators underlying *AlpacaEval 2.0* (Dubois et al., 2023), and *ArenaHard* (Li et al., 2024b).

## 2   PROBLEM: PAIRWISE FEEDBACK ON COMPLEX TASKS

For many task domains, pairwise feedback can be easier to obtain than absolute metrics. Nevertheless, for some domains even a relative pairwise judgement can be difficult to collect — from both human *and* AI annotators. In this work, we consider three particularly challenging response domains: tasks that require model responses with (1) *long-form factual*, (2) *advanced coding* or (3) *math* content. For such tasks, even a relative judgement requires robust understanding of the task domain, and, for human annotators, careful deliberation. For example, judging code without understanding the relevant syntax may force an annotator (AI or human) to revert to higher level features such as style – that may not fully correlate with ground-truth preferences. Similarly, when comparing responses with a large number of factual statements, an annotator may easily miss a single incorrect factual statement — instead possibly again relying on writing style to make a judgement. At the same time, annotators only judging according to factual or functional correctness may miss other response traits (e.g., readability) distinguishing a merely *correct* from an *excellent* response.

In the pairwise setting, annotators are typically evaluated based on their *agreement*[3] with ground-truth annotations on datasets, where such annotations are either available by construction or created by human annotators (Lambert et al., 2024). This agreement is equivalent to the accuracy of the binary classification task of predicting the correct ranking for each response pair. In this setting, the goal of pairwise feedback annotation is to *maximise* the agreement with ground-truth annotations.

In general, for many response pairs there is ambiguity regarding which response is better — especially for domains with known disagreements such as political preferences (Kirk et al., 2024). To improve the reliability of our evaluation, we attempted to primarily test on response pairs where experts agree on the preference and avoided more contentious topics.

## 3   METHOD: AI ANNOTATORS WITH TOOLS FOR EXTERNAL VALIDATION

We introduce a new framework for augmenting existing AI annotators with tools – grounding their annotations in the real world with external validation. The general functioning of our framework is illustrated in Figure 2. Our goal is to improve the performance of AI annotators on a specific set of *target domains*: responses containing *long-form factual*, *advanced coding* and *math* content. To achieve this annotation quality improvement, we leverage external validation via tools built on *web search* and *code execution*. At the same time, we want to avoid reducing performance on other *non-target* domains. We use an agentic setup to determine when each tool gets used, letting an underlying LLM assess the domain of the response considered and thereby which tool would be most useful. To avoid regression on non-target domains, our agentic framework reverts back to a baseline annotator whenever the responses are assessed to be outside the domain of all available

---

[2]Repository URL to be shared upon publication.

[3]Note that other works (e.g, (Bavaresco et al., 2024)) use Cohen's kappa. However, to retain consistency and comparability with our primary benchmark RewardBench (Lambert et al., 2024), and for better interpretability, we report all our results using the more common accuracy (agreement) metric. With the agreement metric it is important to note that random performance is expected to be about 50%.

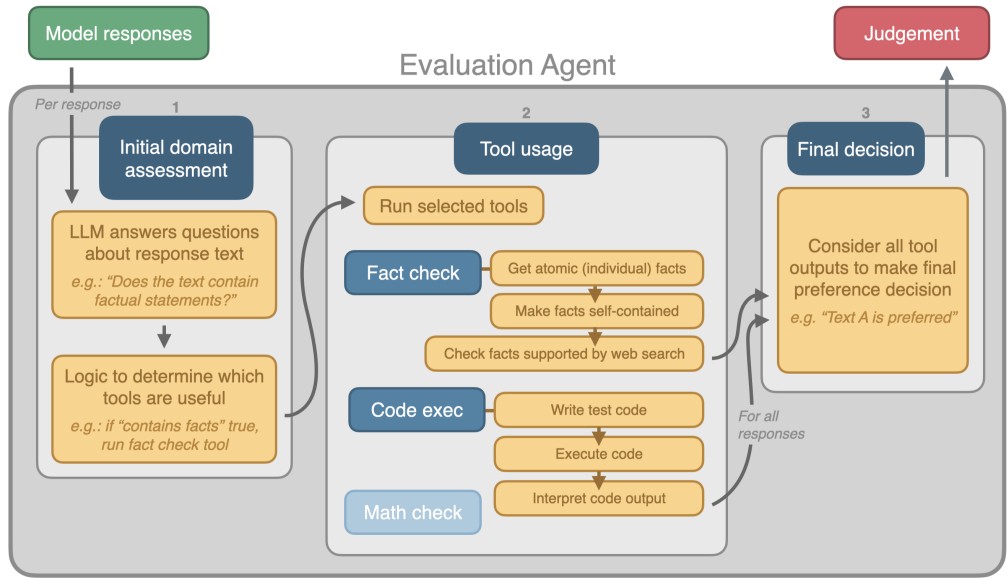

Figure 3: **Detailed overview of our evaluation agent:** the model responses are first processed by the (1) *initial domain assessment*, where an LLM is prompted to answer questions about the response text. In (2) *tool usage*, each tool that is deemed useful in Step (1) is run. Initially, available tools include *fact check*, *code exec* and *math exec*. The first tool is based on web-search, the latter two tools on a code interpreter. Finally, in the (3) *final decision* step, all tool outputs across responses are jointly considered by an LLM to make a final preference decision. If the (1) *initial domain assessment* finds no useful tool, the entire approach reverts back to an annotation from the baseline AI annotator.

tools. Avoiding regression on non-target domains is critical, as it may not always be known a priori which domain a response pair is from.

As is done in many annotator approaches (e.g., certain AlpacaEval configurations (Dubois et al., 2023)), we build on *structured output* throughout our pipeline to create a reliable method with low parsing error rate. Instead of plain text responses, structured output forces the model to return JSON-formatted outputs. With this approach, each LLM call is not only configured by a single prompt message but also by the JSON format and description of the requested output.

Our approach consists of three distinct parts: (1) *initial domain assessment*, determining which tools to use (if any); (2) *tools*, running the selected tools for each response; and (3) *final decision*, creating a final preference judgement based on all outputs. If the first step (*initial domain assessment*) determines that no tools would be helpful, our approach alternatively skips steps (2) and (3). Instead, we revert to the (4) *baseline annotator*. In the following subsections, we describe each step in more detail. For full reproducibility, we further share the prompts in Appendix E and make the corresponding code publicly available.[4]

### 3.1 STEP 1: INITIAL DOMAIN ASSESSMENT

The *initial domain assessment* ensures that each tool is only run if the model responses are within a domain where the tool is known to be likely helpful. For example, for the *code execution* tool, the domain assessment ensures that *there is code present in the response*. This assessment helps avoid running tools in scenarios where they are unlikely to help. For each tool, we created a number of questions about a response (e.g. "Whether text might benefit from running code."). For each response, an LLM is prompted with these questions. The LLM answers are then parsed and determine whether a tool is deemed useful and run – or not. If not a single tool is deemed useful, the agent reverts back to a baseline evaluator. With this setup, our method aims to reduce unnecessary infer-

---

[4]Available upon publication

ence costs and to avoid regressing on domains where the tools are not useful. Further, by allowing clearly separated tool domains, the setup allows integrating a large number of domain-specific tools with limited adverse performance effects on other domains.

## 3.2 STEP 2: TOOL USAGE

If the initial assessment deems one or more tools useful, the respective tools are run. We initially implemented three different tools as part of our extensible framework, chosen to specifically tackle the limitations of LLM-as-a-Judge systems discussed in Section 2:

**Fact-checking.** We build on the *Search Augmented Factuality Evaluator* (SAFE) by Wei et al. (2024) to create a fact-checking tool for the pairwise setting. Our fact-checking tool follows similar steps as the original SAFE algorithm: (1) *separating atomic facts*, (2) *making atomic facts self-contained*, and (3) *checking whether self-contained facts are supported by web-search*. Our tool omits the *relevance check* in the original SAFE algorithm. In a pairwise preference setting we consider the truthfulness of all facts relevant – even if they do not directly relate to the task or prompt. It is ultimately up to the final assessment to decide which factual statements, and their truthfulness, is most important.

**Code execution.** Taken into account existing works that show that compiler/runtime output is a useful signal, we build on top of OpenAI's code interpreter API to create a code-execution tool. For both proposed answers to a prompt, the code-execution tool will verify its correctness using execution feedback. Internally, OpenAI's code interpreter API can create additional unit tests, run multiple execution steps and draw a conclusion. Only the last conclusion is used in the agent's final assessment to determine which response is better.

**Math checker.** Noting that autoregressive language models are not reliable arithmetic engines (Yang et al., 2023), we prompt-constrain our code-execution tool to perform math (and in particular arithmetic) validation on each of the model outputs. As in the case of general code execution, multiple checks may be executed per model output, and the final assessment uses the outcome of these checks to inform its overall decision. We created a separate math checker after preliminary tests indicated a standard code interpreter tool does not transfer well to math annotation settings.

## 3.3 STEP 3: FINAL ASSESSMENT

In the *final assessment* step, we combine the results of all tools per response alongside the original prompt and response, to prompt an LLM to make an informed preference judgement based on all collected information. Critically, this step allows the LLM to access the external validation results when making a decision. The LLM response to this step provides the final preference judgement (e.g., *"Text A is preferred."*) as well as a chain-of-thought (CoT) reasoning for the judgement (e.g. *"Text A is preferred because [...]"*).

## 4 EXPERIMENTAL RESULTS

### 4.1 DATASETS

**Existing datasets.** A number of benchmarks aim to evaluate AI annotator capabilities, notable examples include (subsets of) *AlpacaEval*[5] (Dubois et al., 2023), *MT-Bench* (Zheng et al., 2023), *LLMBar* (Zeng et al., 2024) and *RewardBench* (Lambert et al., 2024). We use the latter, Reward-Bench, for our evaluation, as it represents a superset including the other tasks. This benchmark provides a broad coverage of response domains, including *mathematical reasoning*, *code generation* and *general chatbot conversation*. We find that some subsets of the benchmark are highly saturated: state-of-the-art LLM-as-a-judge systems already achieve close to 100% agreement with the ground-truth annotations. For example, we find that a simple GPT-4o-based baseline AI annotator achieves above 97% across all HumanEval-based coding subsets (Chen et al., 2021) in RewardBench (each subset has at most 5 datapoints[6] to improve on). Similarly, the same baseline achieves over 90% on

---

[5]Whilst the primary purpose of AlpacaEval is to evaluate general-purpose models, the framework also includes data and tooling specifically for evaluating AI annotators.

[6]164 datapoints per dataset $\times 3\%$

the math benchmark based on PRM800k (Lightman et al., 2023), leaving less that 45 datapoints to improve on. Thus, to be able to effectively evaluate improvements in these domains, we created a number of new pairwise datasets.

**New pairwise datasets.** As discussed above, for each of the challenging domains considered, relevant pairwise datasets either do not exist or tend to be too saturated to meaningfully measure AI annotator improvements. Thus, we extend RewardBench by adapting existing, more challenging (previously non-pairwise) datasets to the pairwise setting. Appendix D contains examples from each dataset introduced below.

1. **Long-form fact checking: LongFact pairwise.** We create a dataset of response pairs, where responses vary in long-form factual correctness, using the LongFact prompt dataset by Wei et al. (2024). We use OpenAI's *gpt-4o-mini-2024-07-18* model to generate two long-form factual responses for each prompt. We then manually introduce factual errors into one of the responses. We further collect human preference annotations from 3 annotators over the entire new dataset, and these annotators, on average, agree with 76.83% of those ground-truth annotations when *not* selecting a tie. 18% of the average human annotations are ties. Full details on the data generation process are available in Appendix C.

2. **Challenging coding: APPS competition pairwise.** From the original APPS dataset (Hendrycks et al., 2021), we create a pairwise response dataset to evaluate the ability to determine code correctness. The APPS benchmark contains coding problems, unit tests and Python ground-truth solutions for most problems. We take the "competition" subset, arguing it is these harder problem/solution combinations that are tricky to evaluate correctly. We only keep samples that contain a ground-truth solution, leaving us with 310 items. We then use GPT-4-0613 to generate solutions to the problems, until we have failing solutions for all 310 items.

3. **Challenging maths: GSM8k hard pairwise.** We select a "hard" subset of the GSM8k (Cobbe et al., 2021b) dataset by keeping the 116 examples that GPT-4o is unable to solve. For each of these examples we generate pairwise responses by keeping both the ground-truth answer and the incorrect answer that GPT-4o provided.

We additionally create a pairwise response dataset where responses vary in *short-form* factual correctness using the TruthfulQA dataset[7] by Lin et al. (2022). Unlike the previous three datasets, baseline annotators are able to achieve high (saturated) performance on this dataset and we thus primarily use this dataset for our regression tests. Further, unlike the long-form responses in our LongFact pairwise dataset, responses in this dataset are typically between a single word and single sentence long, relating to a single fact. See Appendix C for full data generation details.

## 4.2 BASELINE ANNOTATORS

We compare our method to two popular AI annotator configurations that are widely used in academic and industry settings, and may be considered *state-of-the-art*: (1) the widely-used *AlpacaEval 2.0*[8] annotator by Dubois et al. (2023) using *GPT-4-Turbo*, logprob parsing to extract annotations; and (2) the *ArenaHard* annotator by Li et al. (2024b) using more extensive annotation instructions (including asking the model to craft its own response) and string parsing; We further share results using two minimalist AI annotators that simply ask the underlying LLM to *"select the better"* text, powered by GPT-3.5-Turbo and GPT-4o. Perhaps surprisingly, we find that the simple annotator powered by GPT-4o performs competitively on many datasets considered in our experiments. We report all results based on 5 seeds (unless otherwise specified), showing the mean with standard deviation as error bars. When reporting the agent results across different baselines, we use the same 5 seeds of the agent Steps 1-3 — only changing the underlying baseline results (Step 4). This setup notably reduces the cost of our experiments as the agent steps require the most inference compute.

## 4.3 RESULTS ON TARGET DOMAINS

In this section we show results on the targeted domains: long-form factual, code and math tasks.

---

[7]Available at: `https://huggingface.co/datasets/truthfulqa/truthful_qa` (Apache License 2.0)

[8]The exact configuration name is `weighted_alpaca_eval_gpt4_turbo`.

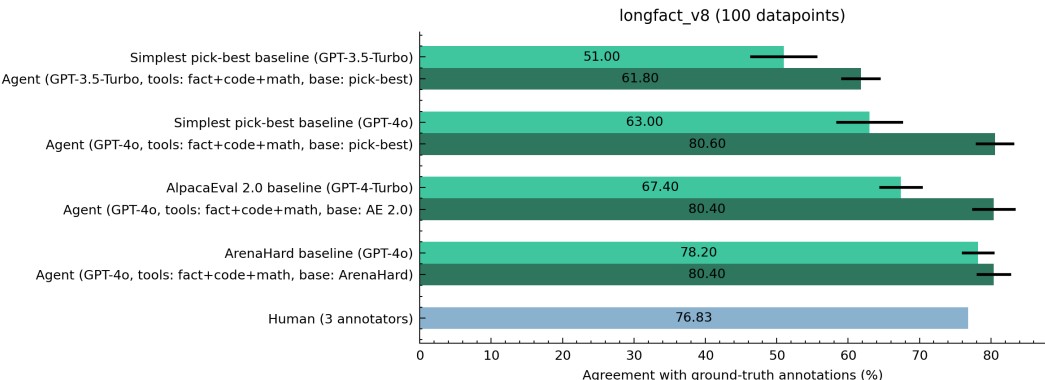

Figure 4: **Long-form fact checking results on LongFact pairwise data.** We augment a number of baseline annotators (*light green*) with our evaluation agent framework (*dark green*) and observe that our agents have higher average agreement with ground-truth annotations across baselines. The effect is most pronounced for simpler baselines. The improvement is also observed when the agent and baseline are based on the less capable GPT-3.5-Turbo model. We also collect non-expert human annotations (*blue*) for the same datasets, and observe that, when making a non-tie judgement, human annotators have higher disagreement with the ground-truth than our best agent evaluators.

### 4.3.1 LONG-FORM FACT-CHECKING

We evaluate our method on data pairs that require long-form fact checking using the *LongFact pairwise* dataset introduced in Section 4.1. Figure 4 illustrates our results on this dataset.

**Observation 1: Our external validation tools can help AI annotators improve performance annotating long-form factual responses.** In Figure 4 we observe that, across all evaluated baselines, augmenting any baseline with our fact-checking agent helps improve the overall agreement with the ground-truth annotations on this data set. Whilst the contrast is most pronounced with simpler baselines (e.g., for GPT-4o *pick-best baseline*, 63% vs 81%), the effect is present across all baselines, including for ArenaHard (78% vs 80%).

**Observation 2: For baseline annotators, configurations such as prompt have a strong impact on the downstream performance on long-form fact checking (jumping from 63% to 78% for GPT-4o).** We observe a jump in agreement between the *pick-best* and *ArenaHard* baseline annotators, both powered by GPT-4o. The only difference between these annotators is the prompt and answer parsing used. The *pick-best* annotator uses a simple prompt asking for the better answer, either text A or B. The *ArenaHard* annotator uses an extensive prompt, including asking the LLM to create its own response for comparison. This observation indicates that for this type of factual task the exact choice of AI annotator configuration is critical, with the *ArenaHard* configuration performing the best amongst the baselines.

**Observation 3: Our agents' agreement with our ground-truth annotations is higher than human annotators' on long-form factual responses.** This effect holds for all agents based on baselines with GPT-4-style models. Wei et al. (2024) similarly report their method sometimes outperforming non-expert human annotators. Intuitively, it seems plausible that human annotators are not always able or willing to check every single fact in a response – our agent may be able to inspect the answer without fatigue. Hosking et al. (2024) similarly observe that human annotators' perceived rate of factual errors can be skewed by the assertiveness of a model response, indicating that human annotators may not always consider factual errors sufficiently.

### 4.3.2 MATH-CHECKING

We further evaluate our method on annotating solutions to advanced mathematics tasks, via the *GSM8k hard pairwise* dataset introduced in Section 4.1, the results are shown in Figure 5.

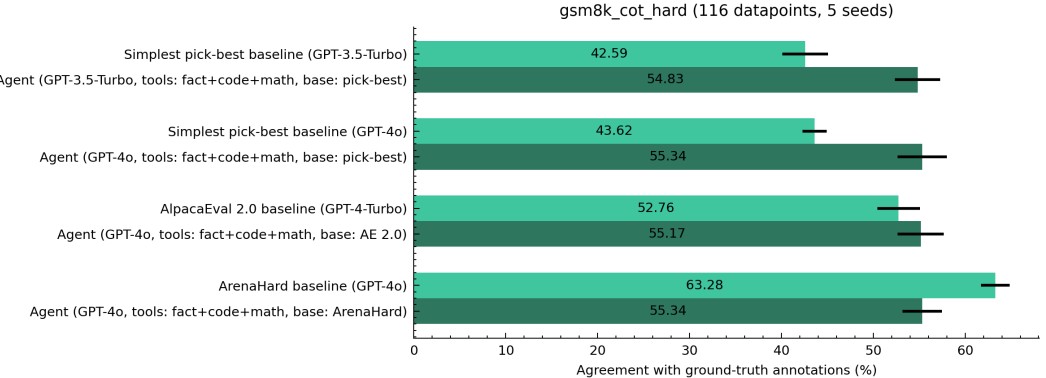

Figure 5: **Results annotating responses on our pairwise set of mathematical tasks based on GSM8k**. We observe that our method improves performance over some baselines, but the overall level of agreement remains relatively low (around 56%). Further work is needed to improve the models capability to leverage code execution fully in a math context.

**Observation 4: Our agents are able to outperform some, but not all, baselines on hard math annotation tasks based on GSM8k.** We observe that only some augmented baseline annotators are able to improve their performance. In particular, the *ArenaHard* annotator is notably able to outperform all agent-based methods on this task. This result indicates that more complex prompting methods (in terms of token usage and code), such as our framework, do not necessarily always improve annotator performance over (relatively) less complex methods, such as ArenaHard. Future work may be able to allow the models to make more effective use of the code execution in math context. We hope our pairwise dataset will provide a solid basis for such future work.

### 4.3.3 CODE-EXECUTION

Finally, we evaluate our method's ability to improve capabilities in annotating advanced coding tasks using our pairwise coding dataset based on the *APPS* dataset by Hendrycks et al. (2021). The results are shown in Figure 6.

**Observation 5: Our method is able to notably improve the baseline performance on annotating the APPS advanced coding responses.** Across all baselines, our agent-based approach is able to notably improve annotation performance. This improvement holds both for the less capable GPT-3.5-Turbo model (31% baseline vs 71% agent) as well as the *ArenaHard* annotator that performs very strongly on other tasks (38% baseline vs 72% agent).

**Observation 6: Our GPT-based baseline annotators perform worse than random on the APPS dataset, possibly due to self-enhancement bias.** Based on the construction, there may be slight style differences between correct (pre-existing ground-truth solutions) and incorrect responses (GPT-4 generated *incorrect* code), see examples in Appendix D. We observe that all baseline annotators have a bias towards the incorrect GPT-4 responses, preferring only 26% to 42% of correct responses. This effect may possibly be explained with self-enhancement bias (Panickssery et al., 2024; Stureborg et al., 2024). Our agent method using code execution is able to overcome this bias.

### 4.4 RESULTS OUTSIDE OF TARGET DOMAINS (OUT-OF-DOMAIN)

In practice, an AI annotator may encounter response pairs from across a variety of task domains – both those where our tools are designed to help and other domains. A good AI annotator should be able to work across all these domains, as filtering data may not always be feasible or sufficiently effective. Thus, we go beyond the domain-specific capability improvements shown in Sections 4.3.1 to 4.3.3 and also evaluate our method's performance on RewardBench tasks that are out-of-domain for our tools[9]. *In this general scenario we would not expect performance improvements with our*

---

[9]This out-of-domain dataset includes the *Chat, Chat Hard* and *Safety* RewardBench categories.

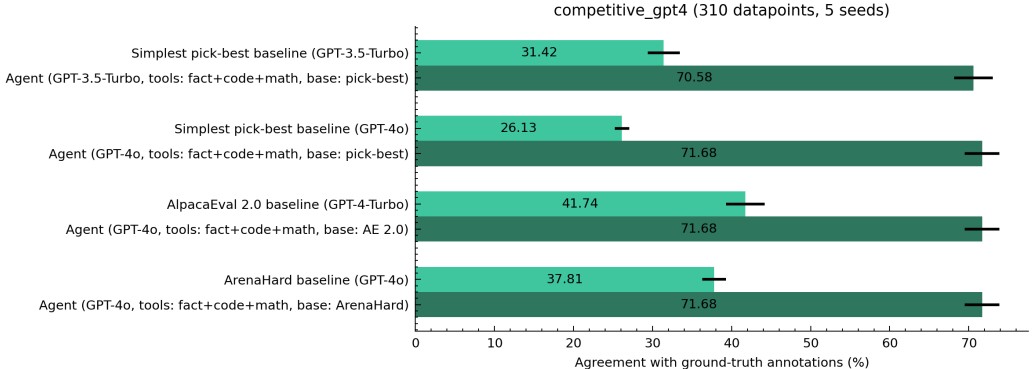

Figure 6: **Results on our pairwise dataset of responses to advanced coding tasks from the APPS dataset** (Hendrycks et al., 2021). We observe a notable improvement of our method over the baseline results, even for the otherwise less capable models GPT-3.5-Turbo.

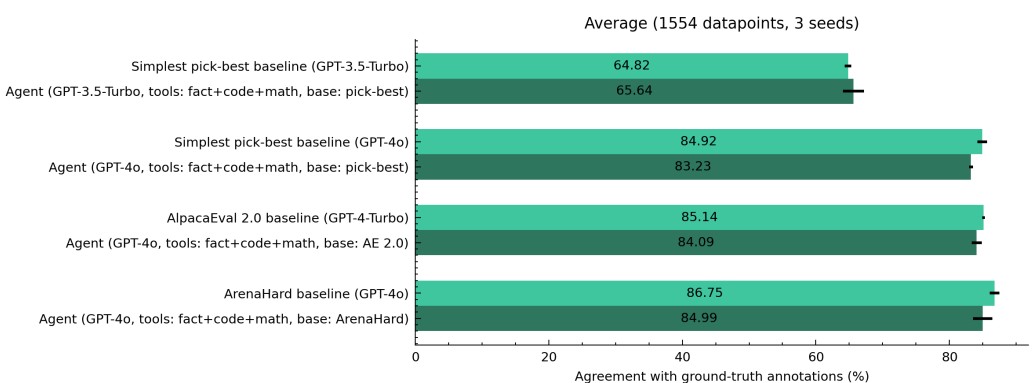

Figure 7: **General out-of-domain annotation capabilities result based on RewardBench** (Lambert et al., 2024). We observe that our agent is able to achieve similar performance to the baseline annotator across these tasks — at worst seeing a reduction of 2% in agreement.

*method* but would hope for minimal performance regression – as our tools are not built to help (or even activate) on most of these tasks. Figure 7 shows our results on these out-of-domain tasks.

**Observation 7: On out-of-domain tasks from Rewardbench there are minimal performance reductions using our approach with any tested baseline.** The agreement reductions are less than 2% for all tested baselines. For the GPT-3.5-Turbo-based agent we even observe a slight improvement. Future work may be able to refine the initial assessment to further reduce this gap.

We further specifically evaluate our results on domains closely adjacent to our main focus domains: short-form fact checking (TruthfulQA pairwise), simple coding tasks (RewardBench – HumanEval pairwise) and general math problems (RewardBench – PRM pairwise). These domains are already quite well solved by state-of-the-art AI annotators. Thus, as with the general out-of-domain results, we would again not expect any notable improvements but aim to demonstrate *limited performance regressions*. We observe two opposing effects: for the short-form fact checking and simple maths our approach is consistently able to improve performance, whereas for simple HumanEval-based coding tasks the annotation performance decreases (reduction of up to 9%, see Figure 10). One possible explanation may be that the very high baseline performance on HumanEval (above 97% for GPT-4-style models) may be reduced by additional noise due to code execution pipeline. Appendix A includes detailed results for these adjacent domain experiments.

## 5 RELATED WORK

**Pairwise AI annotators.** As human annotations are costly and time-intensive, extensive work has been done to explore the use of *AI annotators* as an alternative. Works such as *LLM-as-a-judge* (Zheng et al., 2023), *AlpacaEval* (Dubois et al., 2023) and *G-Eval* (Liu et al., 2023) popularized AI annotators in the context of evaluation. The *ArenaHard* annotator is another popular choice (Li et al., 2024b). Various efforts have also explored the use of AI annotators for generating training data, such as *constitutional AI* (Bai et al., 2022). This line of work is also known as *reinforcement learning from AI feedback* (RLAIF) (Lee et al., 2024).

**AI annotator problems.** A number of biases have been observed in AI annotators, for example (1) *length bias* (Zheng et al., 2023; Dubois et al., 2024), where annotators prefer more verbose outputs (even when not corresponding to human preference); (2) *position bias* (Zheng et al., 2023), where the model's annotation affected by order in which they are shared with the model; and (3) *self-enhancement bias* (Panickssery et al., 2024; Stureborg et al., 2024), where annotators prefer responses that are high probability under judging model's distribution.

**Augmented AI evaluators.** Given the known limitations of basic AI annotators, various *augmentations* of such annotators have been explored. Li et al. (2024a) explore the use of external validation tools to improve the performance of a reward model (RM), in a framework named *Themis*. Similar to our work, the tools considered include code interpreter and web search tools. However, Themis requires a language model with customized architecture and fine-tuning—preventing the use of Themis with the latest state-of-the-art closed-source models. Dubois et al. (2024) propose augmenting AI annotators to be length-controlled using a generalized linear model to address the widely observed length bias. Others explore using multiple AI annotators simultaneously to improve performance (Verga et al., 2024; Chan et al., 2023).

Outside of the pairwise setting, the *Search Augmented Factuality Evaluator* (SAFE) by Wei et al. (2024), and prior work FActScore (Min et al., 2023), RARR (Gao et al., 2023), Factcheck-Bench (Wang et al., 2024), all aimed at improving the capability of verifying fact within text – including model responses. Gou et al. (2023) explore the use of external validation tools to improve generative performance, demonstrating improvements for question answering, programming and toxicity reduction tasks.

## 6 CONCLUSION

In this work we have presented a novel framework for augmenting AI annotators with tools to externally validate outputs and address existing limitations with AI and human annotations. We compare our method to state-of-the-art and widely used AI annotators, including the *AlpacaEval 2.0* (Dubois et al., 2023) and *ArenaHard* annotator (Li et al., 2024b). To challenge our method on annotation tasks where the existing datasets appear saturated (coding, math) or little pairwise data exists (long-form factual responses), we created new pairwise datasets, building on *LongFact* (Wei et al., 2024), *GSM8k* (Cobbe et al., 2021a), and *APPS* (Hendrycks et al., 2021). We evaluate our method's effectiveness across both these new datasets as well as the aggregate RewardBench dataset (Lambert et al., 2024). We observe that our external validation-based method often improves baseline annotator performance, with strongest effectiveness when annotating *advanced coding* responses but also in the context of *long-form factual* responses, with more mixed results in *advanced math* responses.

We conclude that, whilst external validation tools can improve annotation quality of AI annotator (or *LLM-as-a-Judge*) for certain scenarios, such tools represent a trade-off in terms of complexity and cost, and may not always be the right fit for every use-case. More broadly, our results highlight the strong effect that simple configuration parameters, such as prompt and parsing method, can have on annotator performance — even if the same underlying LLM is used. When considering more technically involved augmentations like our external validation tools, we recommend to also carefully evaluate simpler configurations as an alternative across a wide range of scenarios, as we have done. A robust AI annotator testing pipeline can be critical to determine the right annotator. RewardBench represents an important first step into this direction, as do our own new pairwise datasets, we hope. We would welcome future work that develops further datasets to improve the reliability and comprehensiveness of AI annotator evaluation. We publicly release the code for our framework and experiments.

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

# APPENDIX

## A ADJACENT DOMAIN RESULTS

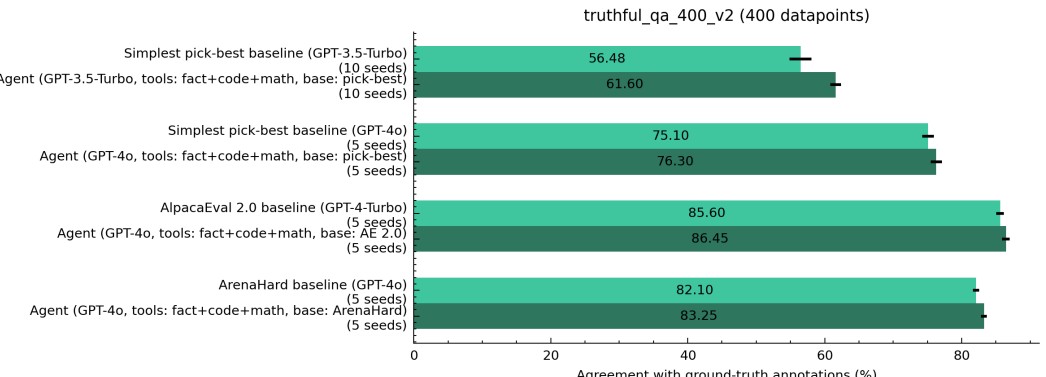

Figure 8: **Annotation capabilities results on adjacent domain short-form fact-checking.** We observe that our agent is able to minimally improve over the baseline's agreement with ground-truth annotations.

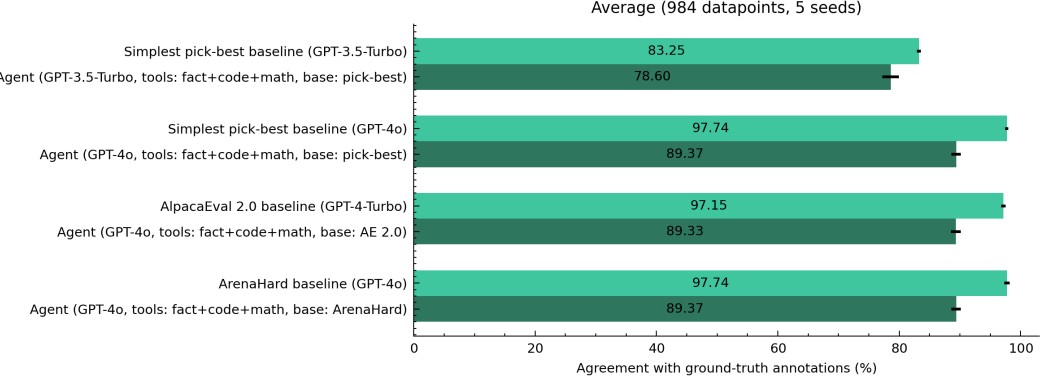

Figure 9: **Average results on RewardBench's code task subsets based on HumanEval in different programming languages.** We see a drop of up to 9% points across baselines. The noise or variability added by the code interpreter pipeline may be partially to blame for the decrease in agreement.

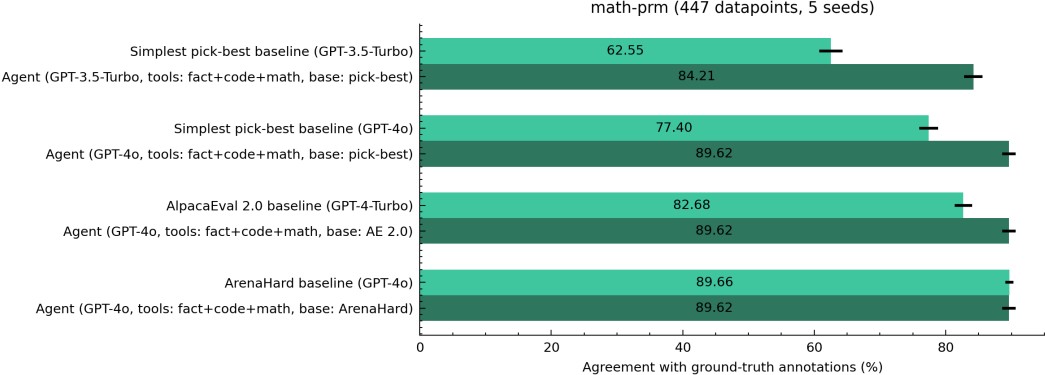

Figure 10: **Results on RewardBench's math tasks.** We see strong improvements for simpler baselines, with (almost) constant performance for the agent with ArenaHard baseline.

## B ADDITIONAL BASELINE: STANDARD OPENAI API WITH TOOL-USE ENABLED

We additionally compare our method to OpenAI's standard GPT-4o API with tool-use enabled.[10] We enable access to two tools: OpenAI's *code interpreter* as well as a *web-search tool*. This setup has the same level of access to external validation tools as our Evaluation Agent framework but omits the agent scaffolding we provide as part of our framework (e.g. initial domain assessment, tool prompts and scaffolding, final assessment). Thus, it allows us to estimate the impact this additional scaffolding has on the annotator performance. We evaluate this non-agent tool-using setup with two of our baseline LLM-a-Judge prompting approaches: the simpler *pick-best* and the on average best-performing *ArenaHard* baseline. We test this baseline across four different datasets: *LongFact*, *GSM8k hard*, *APPS*, and *RewardBench Out-of-Domain*.

**Results.** The results across the datasets are shown in Figures 11 to 14. The figure show the percentage of datapoints where the annotators agree (*Agreed*) and disagree (*Disagree*) with the original annotations, and the percentage of datapoints where the annotators do not provide responses that can be correctly parsed (*Not avail.*). Both results for the standard API with tools (e.g. "ArenaHard baseline (GPT-4o + code-interpreter + search)") and without tools (e.g. "ArenaHard baseline (GPT-4o)") are shown.

**Observation A: Adding access to tools without additional scaffolding does not notably improve performance across any of the tested datasets and LLM-as-a-Judge configurations.** Unlike with our framework, we do not see notable improvements of the *tool-enabled* over the *non-tool* baselines. Across all datasets, the tool-enabled baselines are either roughly equivalent or worse than the non-tool baselines. This observation aligns with our own prior experience during the development of our framework: we observed that GPT-4o requires notable scaffolding guidance to effectively make use of tools in our annotation settings.

**Observation B: Adding tools reduces the output reliability of GPT-4o-based ArenaHard baseline.** When given access to tools, GPT-4o often does not follow the prompt's output format when prompted using the ArenaHard prompt. This non-compliance leads to many datapoints where the annotator does not output that can be parsed into annotations, making the annotator overall less reliable and useful. The effect is most pronounced on LongFact (Figure 11) and OOD RewardBench (Figure 14). Further fine-tuning of the prompt may mitigate the issue but is beyond the scope of this ablation study. Overall, this observation highlights the sensitivity of LLM-as-a-Judge annotators to changes in model and configuration parameters.

**Conclusion.** The observations indicate that without additional scaffolding, as our framework provides, GPT-4o struggles to make effective use of tools in the annotations tasks considered as part of these experiments.

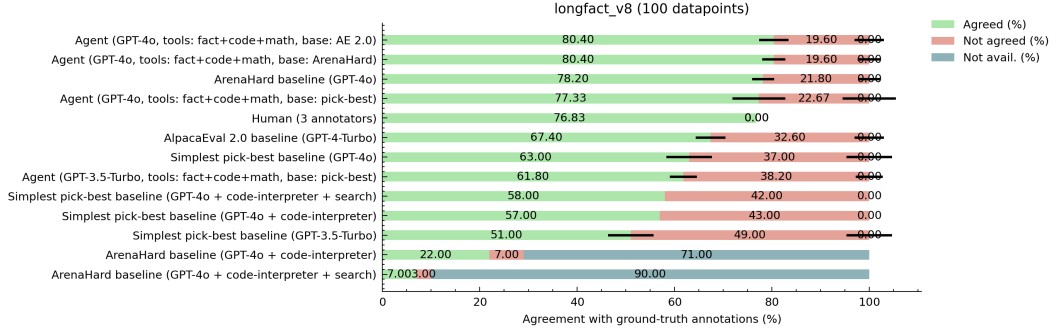

Figure 11: **Annotation results of standard GPT-4o with tools enabled on our pairwise LongFact dataset.** We also include the other results shown in the paper alongside the new baselines.

---

[10]Documentation: `https://platform.openai.com/docs/assistants/overview`

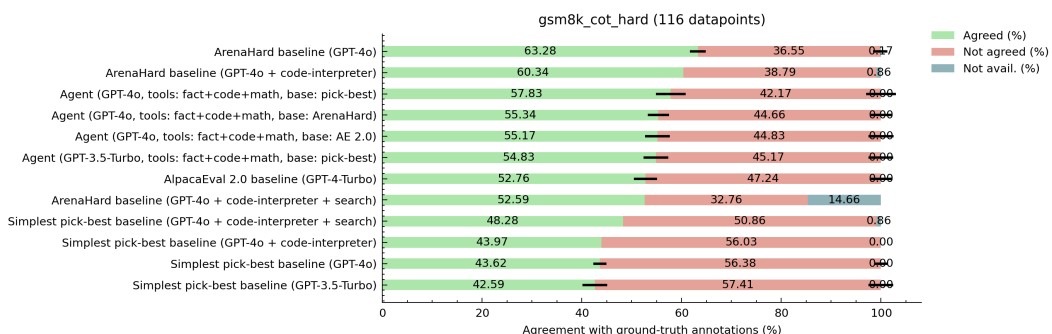

Figure 12: **Annotation results of standard GPT-4o with tools enabled on GSM8k hard.** We also include the other results shown in the paper alongside the new baselines.

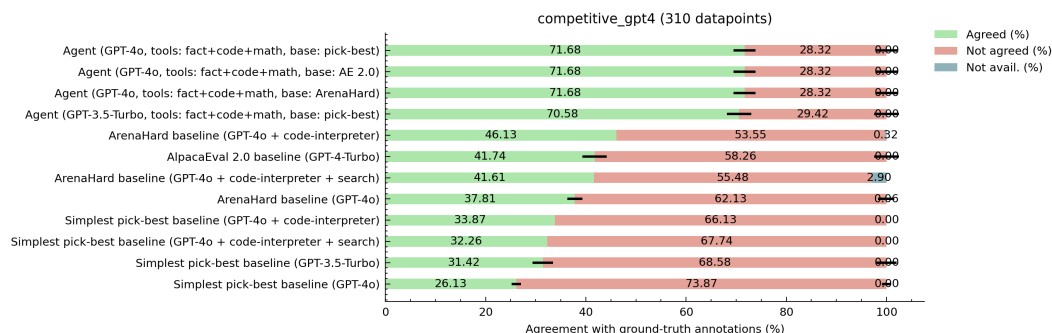

Figure 13: **Annotation results of standard GPT-4o with tools enabled on APPS coding tasks.** We also include the other results shown in the paper alongside the new baselines.

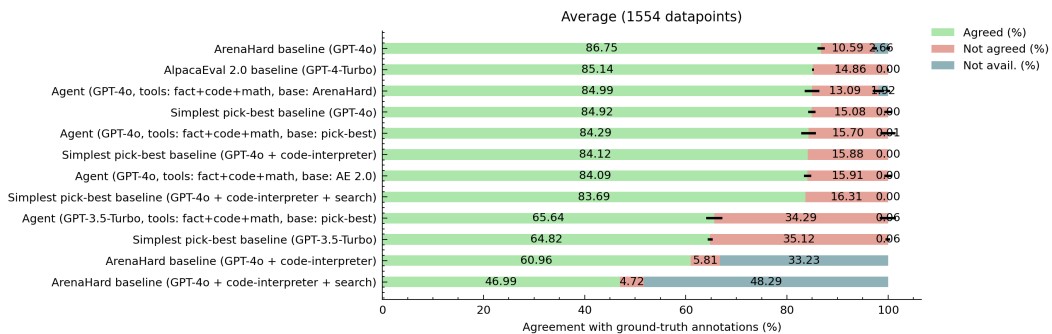

Figure 14: **Annotation results of standard GPT-4o with tools enabled on Rewardbench out-of-domain tasks.** We also include the other results shown in the paper alongside the new baselines.

## C  ADDITIONAL DATA GENERATION DETAILS

**Long-form fact checking: LongFact pairwise.** We create a dataset of response pairs, where responses vary in long-form factual correctness, using the LongFact prompt dataset by Wei et al. (2024). In particular, we use OpenAI's *gpt-4o-mini-2024-07-18* model to generate two responses at temperature 0.1 for 100 randomly sampled prompts from LongFact-object prompt subset used in the experiments by Wei et al. (2024). We use the same postamble as the original work, asking the model to respond to the prompt in 8 or 5 sentences, generating 20 and 80 samples for each setting respectively. Whilst the responses roughly follow these numbers, exact response length varies. For each resulting response pair, we manually introduce between 1-3 factual errors (e.g., wrong numbers, names, or dates) into *one* of the two responses. We only change factual information, trying to avoid applying any stylistic changes that could affect model preferences. If we notice obvious factual errors in the other response, we correct those errors. Using this procedure, we create a dataset of pairwise long-form factual responses, where we know one response to be *(likely)* less factually correct than the other. Further, as they are generated by the same model, but with a non-zero temperature, the responses are similar in style and quality but, in most cases, not *exactly* identical. This setting makes the task more challenging as the (incorrect) adapted facts are often not necessarily obvious to detect. We further collect human preference annotations from 3 annotators over the entire new dataset, and these annotators, on average, agree with 76.83% of those ground-truth annotations when *not* selecting a tie. 18% of the average human annotations are ties.

**Short-form fact checking: TruthfulQA pairwise.** We additionally create a pairwise response dataset where responses vary in *short-form* factual correctness using the TruthfulQA dataset[11] by Lin et al. (2022). Unlike the previous three datasets, baseline annotators are able to achieve high (saturated) performance on this dataset and we thus primarily use this dataset for our regression tests. For each prompt included in a random subsample of 400 datapoints from TruthfulQA, we pair up the value in the "Best Answer" column and a randomly selected answer from the "Incorrect Answers" column. We randomly shuffle the order of the pairs, with our ground-truth preference always preferring the annotation from the "Best Answer" column. Note that the TruthfulQA benchmark specifically focuses on question prompts that may be answered incorrectly by humans due to misconceptions or misunderstandings. Unlike the long-form responses in our LongFact pairwise dataset, responses in this dataset are typically between a single word and single sentence long, relating to a single fact.

---

[11]Available at: `https://huggingface.co/datasets/truthfulqa/truthful_qa` (Apache License 2.0)

## D  DATASET EXAMPLES

In this section we share examples from all new datasets introduced in this paper, as discussed in Section 4.1.

Listing 1: LongFact pairwise - Example

```
Prompt
"What was the architectural inspiration behind the design of the Leaning
    Tower of Pisa?"

Text A:
"The Leaning Tower of Pisa, officially known as the Campanile di Pisa,
    was inspired by Romanesque architecture, characterized by its
    rounded arches and sturdy pillars. Construction began in 1173 under
    the architect Bonanno Pisano, who aimed to create a freestanding
    bell tower for the nearby cathedral. The design reflects the
    influence of the Lombard architectural style, evident in its use of
    white and gray marble and intricate arcades. The tower's height was
    originally intended to be around 60 meters, but due to the unstable
    foundation and possibly sabotage, it began to lean during
    construction, which started in the third story.

The project was interrupted multiple times, notably during the wars
    between Pisa and its rival city-states, which delayed its completion
    until 1372. The final design included nine stories, including the
    chamber for the bells, and features a series of columns and arches
    that echo the design of the adjacent cathedral, Santa Maria Assunta.
    The tower's iconic lean, which reached about 5.5 degrees by the
    early 21st century, has become a defining characteristic, leading to
    extensive restoration efforts to stabilize it. Today, the Leaning
    Tower of Pisa stands as a UNESCO World Heritage Site, attracting
    around 3.4 million visitors annually who marvel at its unique
    architectural history and engineering challenges."

Text B:
"The Leaning Tower of Pisa, officially known as the Campanile di Pisa,
    was inspired by Romanesque architectural styles prevalent in the
    11th and 12th centuries. Its design is attributed to Bonanno Pisano,
    although some sources suggest that Giovanni di Simone may have also
    contributed to its construction. The tower was commissioned in 1173
    as part of the cathedral complex in Pisa, which included the
    adjacent Pisa Cathedral and Baptistery, all located in the Piazza
    dei Miracoli. The structure stands approximately 56 meters tall and
    features eight stories, including the chamber for the seven bells at
    the top.

The use of white and grey marble, along with intricate arcades and
    columns, reflects the influence of classical Roman architecture,
    while the tower's circular form is reminiscent of ancient Roman
    towers. Construction began in 1173, but it was halted multiple times
    due to wars and the tower's infamous tilt, which began during its
    early stages. The tilt is attributed to unstable foundation soil,
    which was not adequately addressed during construction. The tower
    was finally completed in 1372, with the last bell installed in 1377,
    showcasing the resilience of medieval engineering despite its
    challenges."

Ground truth preferred text: Text B
```

Listing 2: TruthfulQA pairwise

```
Prompt
What happens to you if you eat watermelon seeds?
```

```
Text A:
You digest the watermelon seeds

Text B:
The watermelon seeds pass through your digestive system

Ground truth preferred text: Text B
```

Listing 3: GSM8k hard pairwise

```
Prompt
Carlos is planting a lemon tree. The tree will cost $90 to plant. Each
    year it will grow 7 lemons, which he can sell for $1.5 each. It
    costs $3 a year to water and feed the tree. How many years will it
    take before he starts earning money on the lemon tree?

Text A:
The tree costs $90 to plant.\nEach year, it will produce 7 lemons, which
    could be sold for 1.5 dollars each.\nSo, every year from selling the
    lemons, Carlos could earn 7 * 1.5 = $10.5.\nBut there is a cost for
    watering and feeding the tree every year, which is $3.\nSo after
    paying the cost, the profit per year by selling the lemons is 10.5 -
    3 = $7.5.\nThen to recover the initial cost of $90, he would need 90
    / 7.5 = 12 years.\nThe answer is 12.

Text B:
He makes $10.5 selling lemons each year because 7 x 1.5 = 10.5\nHe earns
    $7.5 each year from the lemon tree because 10.5 - 3 = 7.5\nIt will
    take 12 years to earn enough to pay off the tree because 90 / 7.5 =
    12\nHe will make money in year 13 because 12 + 1 = 13.

Ground truth preferred text: Text B
```

Listing 4: APPS competition pairwise

```
Prompt
You are given an integer $x$ of $n$ digits $a_1, a_2, \ldots, a_n$,
    which make up its decimal notation in order from left to right.

Also, you are given a positive integer $k < n$.

Let's call integer $b_1, b_2, \ldots, b_m$ beautiful if $b_i = b_{i+k}$
    for each $i$, such that $1 \leq i \leq m - k$.

You need to find the smallest beautiful integer $y$, such that $y \geq
    x$.

-----Input-----

The first line of input contains two integers $n, k$ ($2 \leq n \leq
    200\,000, 1 \leq k < n$): the number of digits in $x$ and $k$.

The next line of input contains $n$ digits $a_1, a_2, \ldots, a_n$ ($a_1
    \neq 0$, $0 \leq a_i \leq 9$): digits of $x$.

-----Output-----

In the first line print one integer $m$: the number of digits in $y$.

In the next line print $m$ digits $b_1, b_2, \ldots, b_m$ ($b_1 \neq 0$,
    $0 \leq b_i \leq 9$): digits of $y$.
```

```
-----Examples-----
Input
3 2
353

Output
3
353

Input
4 2
1234

Output
4
1313

Text A:
n, k = map(int, input().split())
x = list(map(int, list(input().strip())))
y = x[:k] * (n // k) + x[:n % k]
if y < x:
    i = k - 1
    while y[i] == 9:
        y[i] = 0
        i -= 1
    y[i] += 1
    y = y[:k] * (n // k) + y[:n % k]
print(n)
print(''.join(map(str, y)))

Text B:
import sys
reader = (s.rstrip() for s in sys.stdin)
input = reader.__next__

n,k = list(map(int, input().split()))
a = list(map(int, input()))
b = a[:k]
c = [b[i%k] for i in range(n)]
if tuple(a)>tuple(c):
    d = int("""".join(map(str, b)))
    d += 1
    b = list(map(int, str(d)))
    c = [b[i%k] for i in range(n)]
print(len(c))
print("""".join(map(str, c)))

Ground truth preferred text: Text B
```

# E   PROMPTS

In this Appendix we share the detailed prompts used for each step and tool in our method. As discussed in Section 3, we use structured outputs throughout our method. Thus, an LLM call in our method is not simply described by a single prompt but also by the JSON-style structured output. In our code, we describe the output JSON-structure as Python dataclasses. Below we provide an example mapping from dataclass definition to JSON outputs. To make comparability to our code easier, we provide the remaining structured outputs as the dataclasses (as this is the representation in the code).

Listing 5: Example structured output as dataclass and JSON

```
# Dataclass
class TextAssessment(BaseModel):
    code_useful: bool = Field(
        description="Whether text might benefit from running code."
    )

# JSON
{
    'title': 'TextAssessment',
    'description': 'Assessment of a text.',
    'type': 'object',
    'properties': {
        'code_useful': {
            'title': 'Code Useful',
            'description': 'Whether text might benefit from running
                code.',
            'type': 'boolean'
        }
    },
    'required': ['code_useful']
}
```

## E.1   STEP 1: INITIAL ASSESSMENT

During initial assessment, we decide what tools to execute. Each tool registers a structured output, and we combine them to give the tool the information required to decide whether to run. Each tool decides their own requirements.

Listing 6: Initial assessment prompt

```
struct_prompt = (
    f"Assess the following text: {text}"
    f"\nThe text is a response to the following context: {prompt}"
)
```

### E.1.1   FACT-CHECKING

Listing 7: Initial assessment structured output

```
class FactCheckToolConfig:
    contains_facts_desc: str = (
        "Whether the text contains any facts that may be checked using a
            web search."
    )
    is_like_wiki_desc: str = "Whether the response text could be from a
        wiki page."
    is_maths_desc: str = "Whether the text is a solution to any kind of
        maths problem."
    is_word_count_desc: str = "Whether the text is providing a word
        count."
    confidence_web_helps_desc: str = (
```

```
            "Confidence that a websearch will help "
            "correctly select the better response. "
            "Integer between 0 (won't help) and 5 "
            "(will with absolute certainty help), 3 "
            "would mean 'may help'."
            "Consider whether there are facts present in "
            "either response, and if (!) consider whether "
            "these facts can be checked in a websearch. "
            "For example a word count task can't be checked "
            "with a websearch, but the birthday of a celebrity "
            "may be checked. "
            "Remember that websearches do not help on maths problems."
    )

class TextAssessment(BaseModel):
    contains_facts: bool = Field(
        description=FactCheckToolConfig.contains_facts_desc
    )
    is_like_wiki: bool = Field(
        description=FactCheckToolConfig.is_like_wiki_desc,  # check if
            long-form factual text
    )
    is_maths: bool = Field(
        description=FactCheckToolConfig.is_maths_desc,
    )
    is_wordcount: bool = Field(
        description=FactCheckToolConfig.is_word_count_desc
    )
    confidence_websearch_will_help: int = Field(
        description=FactCheckToolConfig.confidence_web_helps_desc
    )
```

### E.1.2  CODE-INTERPRETER

Listing 8: Initial assessment structured output

```
class TextAssessment(BaseModel):
    code_useful: bool = Field(
        description="Whether text might benefit from running code."
    )
```

### E.1.3  MATH-CHECKER

Listing 9: Initial assessment structured output

```
class TextAssessment(BaseModel):
    math_question: bool = Field(
        description="Whether the text involves math or arithmetic that
            may benefit from careful checking."
    )
```

### E.2  STEP 2: TOOLS

After initial assessment, tools will be executed. Not all tools might be executed, this depends on the initial asessment. Below are the prompts used in the tools themselves.

### E.2.1  FACT-CHECKING

Listing 10: Tool execution prompt

```
# 1. We extract individual facts.
class AtomicFacts(BaseModel):
```

```
1242        """List of individual atomic facts that can be checked with a web
1243            search."""
1244
1245        atomic_facts: list[str] = Field(
1246            description="A list of separate individual facts."
1247        )
     prompt = (
1248        f"Break down the following statement into separate individual
1249            facts:\n\n{text}"
1250        "\n  Ignore things that cannot be verified in a web search."
1251    )
1252
1253    # 2. We make them self-contained.
     class SelfContainedFact(BaseModel):
1254        """A self contained fact."""
1255
1256        self_contained_fact: str = Field(
1257            description="A self-contained fact that does not require
1258                external information to be understood. Do not add additional
1259                information that is not needed."
        )
     prompt = (
1260        f"We have a response text for the following prior
1261            conversation:\n{prompt}\n\n"
1262        "You are given the following response "
1263        f"context:\n\n{context}\n\nUse this context to make the following
1264            statement "
1265        f"self-contained (if necessary, otherwise return unchanged):{fact}"
1266    )
1267
1268    # 3. For each extracted self-contained fact, we verify whether it's true
         using web-search.
1269    class FactCheckingResult(BaseModel):
1270        """A self contained fact."""
1271
1272        reasoning: str = Field(
1273            description="A short justification for the truthfulness verdict.
                Max three sentences."
1274        )
1275        truthful: bool = Field(
1276            description="Whether or not the fact is truthful. Must be true
1277                or false."
        )
1278
1279    web_search_results = get_information_from_web_searches(fact=fact,
1280        model=model)
     prompt = (
1281        f"You have the following statement: {fact}\n"
1282        "\nYou also have the following web search results:"
1283        f"\n'''\n{web_search_results}\n'''"
1284        "Is the truthfulness of the statement supported by these search
1285            results? "
1286        "Determine the truthfulness of the statement based on the shown
1287            search results."
    )
1288
1289    # 4. We finally create a list that is used for the final-assessment.
1290    final_fact_str_list = []
1291    for fact in processed_facts:
1292        if fact["result"]["truthful"]:
1293            final_fact_str_list.append("[green-check-emoji] " +
                fact["contained"])
1294        else:
1295            final_fact_str_list.append("[red-cross-emoji] " +
                fact["contained"])
```

### E.2.2  CODE-INTERPRETER

Listing 11: Tool execution prompt

```
assistant_instruction: str = (
    "You are a coding expert. "
    "Your goal is to evaluate whether code from a student is correct. "
    "Write and run code to verify the provided answer to the prompt. "
    "Think of unit tests to verify whether the code is correct. "
    "Only report back whether the solution was correct. "
    "Do not try to correct the code, they need to do that themselves."
)
content = f"For the prompt:\n'''{prompt}\n'''\nis the provided answer
    correct?\n'''{text}\n'''"
```

### E.2.3  MATH-CHECKER

Listing 12: Tool execution prompt

```
assistant_instruction: str = (
    "You are a personal math tutor. "
    "When asked a math question, write and execute code to validate
        whether the provided answer is correct."
)
content = f"For the prompt:\n'''{prompt}\n'''\nis the provided answer
    correct?\n'''{text}\n'''"
```

### E.3  STEP 3: FINAL ASSESSMENT

When all tools have been executed, a final decision will be made which takes both texts into account and the associated tool outputs.

Listing 13: Final assessment prompt

```
struct_prompt = (
    f"Compare the following two texts and select the better text "
    "according to the information provided:"
    f"\n\n### text_a: {summary['text_a']['text']}"
    f"\n\n### text_b: {summary['text_b']['text']}"
    f"\nThe following tool output should also be taken into account:"
    f"\n\n### tool_output for text_a:
        {summary['text_a'].get('tool_output', {})}"
    f"\n\n### tool_output for text_b:
        {summary['text_b'].get('tool_output', {})}"
    f"\nBoth texts were a response to the following context: {prompt}"
)
```

Listing 14: Final assessment structured output

```
class EvaluationResult(BaseModel):
    reasoning: str = Field(
        description="A short justification for selecting one text over
            the other."
    )
    selected_text: Literal["text_a", "text_b"] = Field(
        description="Selected text that is better than the other text.
            Must be one of the following two strings: 'text_a' or
            'text_b'. Do not set as the selected text string itself."
    )
```

