# OpenReview forum: "Can External Validation Tools Improve Annotation Quality for LLM-as-a-Judge?"
_ICLR.cc/2025/Conference — Submitted to ICLR 2025_

### Official Review · Reviewer_Qxh9 · 2024-10-26

**Soundness:** 2
**Presentation:** 2
**Contribution:** 3
**Rating:** 6
**Confidence:** 4

**Summary:**

This paper investigates whether augmenting LLM-as-a-judge with external validation tools can improve their annotation quality for pairwise feedback. They propose a framework that routes a model response to external tools such as  fact-checking, code execution, and math execution. The outputs of these tools are then collated to inform the final decision of the LLM judge.

To evaluate their proposed framework, the authors constructed benchmarks from existing datasets such as LongFact, APPS, and GSM8k. They measure the **percentage agreement of the LLM judge to the ground-truth annotations of these datasets.** They find significant improvements from baseline annotators on long-form fact checking and coding, but mixed results on math.

The main contributions of this work are as follows:
A framework for augmenting LLM judges with external tools to improve judgments on verifiable / objective domains.
Extension of Rewardbench subsets to create more challenging test sets for fact-checking, coding, and math.

**Strengths:**

- [S1] The contribution is **timely** due to the prevalence of using synthetic preferences from LM judges.
- [S2] The proposed framework is **interesting** as it provides an approach to ground an LLM judge’s annotations to verifiable and objective facts/ground-truth, using existing and off-the-shelf tools today.
- [S3] I also appreciate the effort to **extend subsets of RewardBench to create more challenging test sets** due to the saturation of the said benchmark.

**Weaknesses:**

- [W1] One con for this work is the **insufficiency of experiments to show the accuracy / reliability of specific components of the framework.** For example, how reliable is the “Initial domain assessment” component for routing responses to specific tools?
    - [W1.1] In addition, showing the robustness of the framework as new tools are added to the agent can help strengthen the use-case of this framework.

- [W2] **Lack of motivation** as to why the specific tools (SAFE, OpenAI code, OpenAI math) were chosen for each component. Were there any other components tested? How sensitive are the reported results to these tools?

- [W3] The are some **claims that have shallow to no evidence** (a few notable examples):
    - Section 4.3.2 (Observation 4): The claim is that complexity (e.g., in the form of tools) does not always yield better results. The only evidence so far is ArenaHard outperforming the agent framework, but we also see that other simpler methods like pick-best and AlpacaEval underperformed against the agent framework. Perhaps there are other confounders, and there’s a need to disentangle what complexity means.
    - Section 4.3.3 (Observation 6): There is a claim that baseline annotators have bias towards incorrect GPT-4 responses, and it was explained as self-enhancement bias. It was further claimed that the agent framework’s code execution path overcame this bias. The only evidence so far is the empirical results, but how much of this was due to the code-execution tool and how much was from AlpacaEval (baseline annotator)?
    - Finally, I think it’s important to show how each component contributed to the performance of the overall framework. For the strongest results (Math and Fact-checking), how much of the performance is attributed to the tool and how much was from AlpacaEval?

**Questions:**

Questions
- [Q1] Were there any tests on non-benchmark preference training datasets (e.g., Anthropic-HH, Helpsteer2, ChatArena), and the effect of the agent framework on the downstream reward model / policy model performance?

Comments/Suggestions (these are nits that don’t weigh a lot in my scoring but I’d appreciate if addressed as it can improve the manuscript):
- [C1] There are some non-formal words used throughout the text that I would appreciate if corrected:
Page 6, bullet point #2, last sentence: “till we have failing solutions” -> “until…”
- [C2] The term agentic was introduced suddenly in p.2 without any introduction / contextualization as to what it means.

---

> ### Author Response · Authors · 2024-11-25
>
> Thank you for the well-structured review! Please find below our responses to each of the weaknesses (W) and questions (Q) raised in your review. All references refer to the updated manuscript, where we highlight relevant changes in blue.
>
> > **W1: One con for this work is the insufficiency of experiments to show the accuracy / reliability of specific components of the framework. For example, how reliable is the “Initial domain assessment” component for routing responses to specific tools?**
>
> The different datasets from varying domains aim to evaluate the reliability the individual tools (i.e., LongFact for the fact check, GSM8k for math check, and APPS for the code check tool).
>
> However, we agree that more could be done to evaluate the impact of the other parts of the evaluation agent system, especially the initial domain assessment component. To test the impact of the other components on reliability, **we run and share another set of experiments where we remove the "agentic" components** (initial domain assessment, final decision and in-tool scaffolding) and simply use OpenAI's assistant API with a code interpreter and search tool. We test adding tools in this way to two of our baseline LLM-as-a-Judge methods: the simple pick-best and overall best-performing ArenaHard baseline. We the following observations: (1) adding access to tools without additional scaffolding does not notably improve performance of the baselines across any of the tested datasets and LLM-as-a-Judge configurations (often it decreases performance); and (2) adding tools reduces the output reliability of GPT-4o-based ArenaHard baseline.
>
> We consider these new observations a strong indication that the non-tool components of our framework help the reliability of our system. See a more detailed discussion of these results in Appendix B of our updated manuscript.
>
> > **W1 (1): In addition, showing the robustness of the framework as new tools are added to the agent can help strengthen the use-case of this framework.**
>
> The effect of adding new tools on the robustness of the framework depends on two factors: the configuration of the initial domain assessment and the impact of the tool on annotation performance when activated. For each tool, the initial domain assessment determines if the given annotation task is *in-domain* for that tools, by asking an LLM a series of questions about the annotation task. By limiting domains where the tool activates to domains where we have high confidence of the tools efficacy, we improve robustness. Additional care is required when two tools activate on the same annotation domain. Then, we recommend running ablation experiments to validate that joint tool usage is more effective than single-tool usage. Thus, the effect of adding a new tool can be controlled using these configuration parameters.
>
> > **W2 (1): Lack of motivation as to why the specific tools (SAFE, OpenAI code, OpenAI math) were chosen for each component. Were there any other components tested?**
>
> Thank you for raising this point! We have clarified our motivation in Section 3 of the updated manuscript but also summarize this clarification as well as more details on the development process below.
>
> Based on the chosen annotation domains (long-form factual, math and code tasks), we initially selected two categories of tools to evaluate: fact-checking tools and code-interpreter-based tools. For fact-checking, we quickly determined the SAFE method to be effective. Our development effort primarily focused on determining the domains where fact-checking works well. For math and code tasks, we initially tested a single code-interpreter tool but found it to not work well across code and math tasks. Thus, instead we created two separate code-interpreter-based tools with custom prompts and domain assessment, one for code tasks and one for math tasks. We updated Section 3 of our manuscript to better reflect this motivation.
>
> > **W2 (2): How sensitive are the reported results to these tools?**
>
> Generally, as our baseline experiments and the AlpacaEval annotator leaderboard [1] indicate, LLM-as-a-Judge systems can be very sensitive to the specific prompting setup used --- even if the same base model is used (e.g. GPT-4o). In our experience, this sensitivity also applies to tool-use within our framework. As such, each tool needs to be carefully configured in terms of domain and output to ensure that the provided information is effectively used by the model.
>
> [1] https://github.com/tatsu-lab/alpaca_eval/tree/main/src/alpaca_eval/evaluators_configs

---

> > ### Author Response · Authors · 2024-11-25
> >
> > > **W3: The are some claims that have shallow to no evidence (a few notable examples):**
> >
> > > **W3 (1) Section 4.3.2 (Observation 4): The claim is that complexity (e.g., in the form of tools) does not always yield better results. The only evidence so far is ArenaHard outperforming the agent framework, but we also see that other simpler methods like pick-best and AlpacaEval underperformed against the agent framework. Perhaps there are other confounders, and there’s a need to disentangle what complexity means.**
> >
> > Thank you for raising this point! We hope to be able to clarify the intended meaning here, and adjusted the corresponding section for better clarity.
> >
> > We refer to the complexity of the LLM-as-a-Judge prompting strategy, in terms of amount of used input/output tokens as well as the time requirements of the surrounding code (e.g. web-search, code interpreter). In this interpretation of complexity, our simple pick-best baseline is our "least complex" method tested (minimal input/output tokens, minimal surrounding code), whilst our evaluator agent with all tools the "most complex" (many input/output tokens, extensive surrounding code). ArenaHard's complexity is in-between as it uses more tokens than the pick-best baseline, but less than our agent and requires simpler surrounding code (no web-search, code interpreter).
> >
> > Across our experiments, we observe that the more complex LLM-as-a-Judge methods (according to the above definition) win often, but not always. For example, (1) on GSM8k hard (Fig. 5), the "simpler" ArenaHard method outperforms our agent; or (2) on APPS coding (Fig. 6) the "simpler" AlpacaEval 2.0 method outperforms ArenaHard.
> >
> >
> > > **W3 (2) Section 4.3.3 (Observation 6): There is a claim that baseline annotators have bias towards incorrect GPT-4 responses, and it was explained as self-enhancement bias. It was further claimed that the agent framework’s code execution path overcame this bias. The only evidence so far is the empirical results, but how much of this was due to the code-execution tool and how much was from AlpacaEval (baseline annotator)?**
> >
> > As you state, we observe that the GPT-4o-based baseline annotators have bias towards the incorrect GPT-4 responses. In Observation 6, we suggest self-enhancement bias as a *"possible"* explanation, as the incorrect responses were generated by the same model family, and self-enhancing behavior (a model preferring its own outputs) has been observed as a possible explanation in similar situations [1][2]. However, we cannot rule out that there might be another bias or effect involved. We have rephrased the observation to make it clearer that self-enhancement bias is suggested as a *possible* rather than directly observed explanation.
> >
> > With respect to the agent framework overcoming this bias, this effect can generally be fully attributed to the agent, as the underlying baseline annotator has exactly the same bias. However, we agree that it is interesting to investigate how the bias is on the remaining annotations, annotated when the agent reverts to the baseline. We are currently working on extracting these metrics and will update our response in due course.
> >
> > - [1] https://arxiv.org/abs/2404.13076
> > - [2] https://arxiv.org/abs/2405.01724
> >
> > > **W3 (3) Finally, I think it’s important to show how each component contributed to the performance of the overall framework. For the strongest results (Math and Fact-checking), how much of the performance is attributed to the tool and how much was from AlpacaEval?**
> >
> > We are currently working on extracting these metrics and will update our response in due course.
> >
> > > **Q1: Were there any tests on non-benchmark preference training datasets (e.g., Anthropic-HH, Helpsteer2, ChatArena), and the effect of the agent framework on the downstream reward model / policy model performance?**
> >
> >  We intentionally excluded such training datasets, as we could not necessarily rule out that the language models used in our experiments have been trained on these datasets, potentially triggering misleading results. The authors of RewardBench [1] explicitly highlight that "substantial research is needed to understand the full limitations of these previous datasets", referring to training preference datasets, including Anthropic-HH.
> >
> >  - [1] https://arxiv.org/abs/2403.13787

---

> > > ### Author Response · Authors · 2024-11-25
> > >
> > > > **C1: There are some non-formal words used throughout the text that I would appreciate if corrected: Page 6, bullet point #2, last sentence: “till we have failing solutions” -> “until…”**
> > >
> > > Thank you for highlighting this use of informal language, we corrected this.
> > >
> > > > **C2: The term agentic was introduced suddenly in p.2 without any introduction / contextualization as to what it means.**
> > >
> > > Thank you for highlighting this omission. Given that the term "agent" is used with various meaning in the literature, we agree that this term needs a more precise introduction. We have now rephrased the introduction to *"Our framework is agentic in the
> > > sense that an LLM assesses the response domain and plans the optimal tool usage accordingly"*. Let us know if you think further clarification would be helpful!

---

> > > > ### Author Response · Authors · 2024-12-02
> > > >
> > > > Thank you again for taking the time to review our work! With the discussion period ending soon (less than a day for reviewer comments), we wanted to briefly follow up to ask if there are any remaining concerns or questions regarding our response above?

---

> ### Comment · Reviewer_Qxh9 · 2024-12-02
>
> Thank you for these clarifications and for updating the manuscript. As it stands, the manuscript has improved. I have also read the other reviews and agree with Reviewer Z3LQ on his concern about novelty. However, I appreciate the systematic approach the manuscript has done in testing these different tools. I have adjusted my rating to 6.

---

> > ### Author Response · Authors · 2024-12-04
> >
> > Thank you for taking the time to consider our response and revised manuscript, and updating your score! We appreciate your help in improving our paper!
> >
> > Final update for completeness: With respect to the performance attribution metrics (W3(3)), we were unable to complete this analysis in time due to time constraints and the complexity of the other experiments, but would aim to add this analysis to a camera-ready version.

---

### Official Review · Reviewer_jhb8 · 2024-11-02

**Soundness:** 3
**Presentation:** 4
**Contribution:** 4
**Rating:** 8
**Confidence:** 4

**Summary:**

This paper introduces the concept of external validation of ground truth information during pairwise judgements -- that is, when judging pairs of responses to a given prompt, before using an automated annotator to judge which response is better, tools are first used to validate pieces of the outputs (code execution & correctness, mathematical reasoning, and factuality) and this information is then provided as additional information to the annotator. On datasets with ground truth information (i.e. pairs where one is confirmed to be better than the other), their method noticeably improves performance on factuality and coding tasks, and has less clear performance gains on math tasks.

**Strengths:**

This paper proposes a reasonable and interesting framework for improving pairwise judgements using automated annotators. Recent work has shown the strength of strong, automated pairwise annotators, and this work is a valuable extension of that, showing that ground truth information in the responses (that traditional LLM-only systems might not always pick up on) is valuable for making these decisions.

**Weaknesses:**

While this paper shows strong results on annotation accuracy, it is unclear how well this improves downstream performance. I don't think this is a hard requirement for this work, but I'd be interested to see how model performance changes using this method to either generate preference data, or do best-of-n ranking for model outputs. I do not think this is required for this paper to be accepted, however.

**Questions:**

I have one question, and one suggestion:
* Did you check the correctness of the GSM8K hard answers? GSM8K has a small but noticeable subset (<5%) that have incorrect labels, so without any validation, the instances that GPT4o gets "wrong" may be mislabeled. I'd recommend checking this, and if some are mislabeled, this may be the source of the mixed results you see on math reasoning. If so, I'd recommend thinking about harder math datasets (like MATH), though this may be more complicated for code execution.
* I'd be interested to see how this affects best-of-n ranking when using LLMs as a judge for ranking n model outputs -- I'd assume this would noticeably help performance on the domains tested. This may be expensive depending on the setup though, so this is also a reasonable follow up work instead.

---

> ### Author Response · Authors · 2024-11-25
>
> Thank you for your constructive review, especially for highlighting potential issues with GSM8k! Please find below our responses to each of the weaknesses (W) and questions (Q) raised. All references refer to the updated manuscript, where we highlight relevant changes in blue.
>
> > **W1: While this paper shows strong results on annotation accuracy, it is unclear how well this improves downstream performance. I don't think this is a hard requirement for this work, but I'd be interested to see how model performance changes using this method to either generate preference data, or do best-of-n ranking for model outputs. I do not think this is required for this paper to be accepted, however.**
>
> We agree that looking at effects on downstream performance would be a very interesting direction for future work! We consider the experiments included in our paper, that are based on the annotation of diverse set of preference datasets, as the most direct evaluation of our method's capabilities. Yet, testing downstream fine-tuned model performance would help obtain a more comprehensive understanding of the effects of our method. Unfortunately, given the limited time available, we were unable to conduct such a more resource-demanding experiment. We would be excited to see future work explore the downstream use-cases, both in terms of model fine-tuning and evaluation.
>
>
> > **Q1: Did you check the correctness of the GSM8K hard answers? GSM8K has a small but noticeable subset ($<5\%$) that have incorrect labels, so without any validation, the instances that GPT4o gets "wrong" may be mislabeled. I'd recommend checking this, and if some are mislabeled, this may be the source of the mixed results you see on math reasoning. If so, I'd recommend thinking about harder math datasets (like MATH), though this may be more complicated for code execution.**
>
> Thank you for highlighting this potential issue! We had previously manually checked *some* of the datapoints, but following your suggestion **we have done a thorough manual check of the validity of all GSM8k hard datapoints**.
>
> **Process.** We first compared our results to errors in GSM8k that were publicly reported [1][2]. We found two incorrect datapoints included, approx. $2\%$ of the dataset. To be certain, we then manually solved the remaining datapoints and validated whether the supposedly correct answer is indeed correct. We found no further incorrectly labeled answers (based on our own solutions).
>
> **Results.** Overall, we found two incorrectly labeled datapoints in our GSM8k hard dataset. For both samples, we observe that our agent models consistently prefer the (actually correct) GPT-4o generated datapoints (rather than the incorrect golden perferences), whereas the baseline models only sometimes prefer the golden datapoints. This effect indeed, as you suggested, may slightly inflate the performance of baseline models, but by less than $2\%$. Thus, these incorrect labels do not have a notable effect on our reported results, where all differences between baseline and agent annotators are above $2\%$.
>
> Further, we note that as part of this inspection we also found a small typo in our dataset description (L299): our GSM8k hard dataset consists of 116 rather than 117 datapoints. This error has been corrected in the updated manuscript and does not affect our results.
>
> Overall, thank you again for helping us make these results more robust!
>
> - [1] https://huggingface.co/datasets/Cleanlab/bad_data_gsm8k_svamp.csv/
> - [2] https://github.com/openai/grade-school-math/issues
>
>
> > **Q2: I'd be interested to see how this affects best-of-n ranking when using LLMs as a judge for ranking n model outputs -- I'd assume this would noticeably help performance on the domains tested. This may be expensive depending on the setup though, so this is also a reasonable follow up work instead.**
>
> See our response to W3. We further add that we agree that we would also expect the impact on downstream model performance to be primarily visible in similar task domains as covered in our current experiments. As before, we would be very excited to see this line of investigation in future work.

---

### Official Review · Reviewer_Z3LQ · 2024-11-03

**Soundness:** 2
**Presentation:** 3
**Contribution:** 2
**Rating:** 3
**Confidence:** 4

**Summary:**

This paper proposes a tool-augmented (i.e., web search engine and code compiler) method to provide pairwise AI feedback, with a focus on three specific domains: long-form factual, math, and coding tasks. Specifically, for each incoming pairwise response, it would first determine its domain and then select the corresponding tool for quality judgment. Experiments are conducted on three pairwise datasets sourced from LongFact, APPS competition subset, and GSM8K hard subset. Results indicate that by tool-augmentation, AI feedback improves in most, but not all, cases on these three subsets. While on a general pairwise benchmark Rewardbench, AI feedback slightly decreases.

**Strengths:**

1. clear paper writing
2. Classifying the input domain and selecting tools accordingly makes sense.
3. Substantial improvements on certain subsets, particularly APPS.

**Weaknesses:**

1. My main concern is novelty. Several highly related (i.e., tool-augmented AI feedback), published papers have not been cited and clearly discussed ([1][2]). "Novel framework" sounds overclaim.
2. Studying pairwise feedback in domains with clear objective correctness (e.g., fact, code, math) is unjustified.
3. Mixed results. Performance slightly decreases on general domains (rewardbench) and math when the base model is stronger (e.g., GPT-4o).

[1] https://arxiv.org/pdf/2310.01045
[2] https://openreview.net/pdf?id=Sx038qxjek

**Questions:**

1. Why choose a subset from GSM8K rather than selecting more general benchmarks (e.g., AIME, MATH)?
2. Can you also present experiment results on open-source models?

---

> ### Author Response · Authors · 2024-11-25
>
> Thank you for your in-depth review! Please find below our responses to each of the weaknesses (W) and questions (Q) raised. All references refer to the updated manuscript, where we highlight relevant changes in blue.
>
>
> > **W1: My main concern is novelty. Several highly related (i.e., tool-augmented AI feedback), published papers have not been cited and clearly discussed. "Novel framework" sounds overclaim.**
>
> Thank you for bringing this relevant work to our attention, we agree that this work is related to ours and have added a corresponding discussion to our related work section (Section 5). However, we also emphasize that there are notable differences between the referenced work and our framework. Below we provide a more detailed discussion of the referenced works and how they differ from our work:
>
>
> **Themis.** The *Themis* framework introduced by [Li et al. (2023)](https://arxiv.org/abs/2310.01045) investigates the integration of tools into reward models. Similar to our work, this framework aims to give AI annotators access to a number of external validation tools to improve their annotation quality, including code interpreter and web search tools. __The critical difference to our work is the *modeling choice*__: the Themis approach by Li et. al (2023) combines generative language and conventional scalar reward outputs to create a tool-using reward model. Their setup requires a custom model with separate final layers for language- and reward-modeling. Such an approach *requires* customizing a model's architecture and fine-tuning a model. Our approach, on the other hand, following the general LLM-as-a-Judge direction, works with any generative language model without need for fine-tuning, including models only accessible via APIs. **Our method not requiring fine-tuning has two main benefits:** **(a) easier and cheaper deployment** compared to custom fine-tuned models, which are typically more expensive to train and deploy in terms of development time and compute cost; and **(b) ability to use state-of-the-art closed-source LLMs** that cannot be customized or fine-tuned as required for Themis. For example, OpenAI's `GPT-4o` or Anthropic's `Claude-3.5-Sonnet` could not be used directly with the Themis framework. We consider the ability to use state-of-the-art models critical when trying to annotate the kind of advanced tasks that our paper considers. Further, it is worth noting that anybody with the relevant API key(s) (model and search API) can use our framework with minimal development effort.
>
> Beyond the general modeling differences, the code interpreter implementation in Themis also differs notably from ours: instead of working with any preferences pair containing code, Themis' code interpreter additionally requires the availability of a list of unit tests. This detail is shown in Figure 9 of the Themis paper and we confirmed this in their public implementation. Whilst such tests may be available for certain benchmark tasks, this is not generally the case and limits the applicability of this tool to new data. Our code interpreter tool enables the model to write such tests itself, and thus can be used with new data where such tests are unavailable.
>
> **CRITIC.** The *CRITIC* framework by [Guo et al. (2023)](https://openreview.net/pdf?id=Sx038qxjek) aims to use tool-based self-critiques to improve model outputs. Similar to our work, this framework considers the use of external validation tools to improve model outputs, including web-search and code-interpreter tools. However, CRITIC is intended to directly improve generated responses rather than providing general (pairwise) feedback that can be used for downstream training or evaluation. As such, it is more similar to general prompting techniques like Chain-of-Thought (CoT) [1], rather than LLM-as-a-Judge. Experimentally, the method is validated on general generation tasks such a free-form question answering, mathematical program synthesis, and toxicity reduction. The method is compared to general prompting technique baselines like CoT rather than more specific LLM-as-a-Judge approaches. In general, this work provides valuable information about integrating external validation tools as part of a prompting strategy but cannot be directly compared to our framework as it is intended for general response generation and refinement rather than for providing feedback in an LLM-as-a-Judge setting.
>
> - [1] https://openreview.net/forum?id=_VjQlMeSB_J

---

> > ### Author Response · Authors · 2024-11-25
> >
> > > **W2: Studying pairwise feedback in domains with clear objective correctness (e.g., fact, code, math) is unjustified.**
> >
> > We disagree that studying pairwise feedback for factual, coding or math responses is unjustified. Whilst we agree that for these tasks a notion for objective correctness exists, there are also a number of less easily defined considerations when comparing two responses in these domains: for example readability and style. An excellent model response in these domains should *combine all these properties*. Conventional benchmark methods that only validate the correctness of responses (e.g. HumanEval), do not account for more subtle but nevertheless important differences in responses, such as style and readability. On the other hand, conventional LLM-as-a-Judge systems *are* able to account for these more subtle differences but struggle to correctly account for the functional correctness of the responses. Our framework aims to combine these two approaches: giving LLM-as-a-Judge information about the functional correctness, such that the feedback can account for both the objective correctness as well as more subtle response features such as style and readability. In practice, a good model in any of these domains should produce responses that combine objective correctness with excellent readability and style. Our method may help provide the feedback to get the model there, without compromising on functional correctness. Following your question, we have adjusted Section 2 in the manuscript to state this motivation and justification clearly.
> >
> > > **W3: Mixed results. Performance slightly decreases on general domains (rewardbench) and math when the base model is stronger (e.g., GPT-4o).**
> >
> > Within this work, in line with good academic practice, we aim to provide a realistic and honest assessment of our approach: we intentionally include strong baselines (that sometimes outperform our method) as well as challenging datasets (that are not necessarily favorable to our approach). We consider the observation that our method does not always outperform other methods an indication that our baselines and datasets were well-chosen, rather than a failure of our approach.
> >
> > In general, the task of providing pairwise feedback across natural language domains is highly diverse --- such that it is rare to see universal improvements across all domains. Indeed, we observe that diversity among our baselines: AlpacaEval 2.0 performs best on APPS coding (Figure 6) but performs almost equivalent to the pick-best baseline on out-of-domain RewardBench (Figure 7). The ArenaHard baseline leads on the latter, as well as maths (Figure 5) and fact-checking (Figure 4).
> >
> > Overall, we believe our results indicate that tool-use is a promising direction to improve LLM-as-a-Judge methods but challenges remain to ensure this approach does not degrade performance out-of-domain. We hope our open framework can provide a good starting point for further research in this direction --- based on a realistic assessment of the current capabilities.
> >
> >
> > > **Q1: Why choose a subset from GSM8K rather than selecting more general benchmarks (e.g., AIME, MATH)?**
> >
> > Great question! We also considered the filtered subset of PRM800k [1] included in RewardBench (referred to as *math-prm*). This dataset is based on tasks from the MATH dataset [2]. We found that the strongest baselines already perform very strongly on this dataset: our ArenaHard baseline achieves close to $90\%$. Our agent framework either improves (pick-best, AlpacaEval 2.0) or approximately maintains (ArenaHard) the performance of all baselines on this dataset. We include the results on this dataset in Figure 10 in Appendix A.
> >
> > However, given this high (almost saturated) baseline performance, we looked for other harder maths datasets, which led us to GSM8k hard --- where the best baseline performance is much lower ($~63\%$). We further took into account that a code interpreter is not necessarily equally useful for all maths tasks, but perhaps more so for arithmetic rather than theoretical proofs. Due its make-up, GSM8k hard is challenging for language models but performance may be helped with a code interpreter-based tool.
> >
> > - [1] https://github.com/openai/prm800k
> > - [2] https://github.com/hendrycks/math
> >
> > **Q2: Can you also present experiment results on open-source models?**
> >
> > We will come back to this question in a future response.

---

> > > ### Comment · Reviewer_Z3LQ · 2024-11-27
> > >
> > > > Our method may help provide the feedback to get the model there, without compromising on functional correctness. Following your question, we have adjusted Section 2 in the manuscript to state this motivation and justification clearly.
> > >
> > > just checking: do the current experiments evaluate "subtle differences" beyond correctness?
> > >
> > > > challenges remain to ensure this approach does not degrade performance out-of-domain.
> > >
> > > I'm focusing on the degradation in iid setting (Figure 5) instead of ood.
> > >
> > > > We found that the strongest baselines already perform very strongly on this dataset: our ArenaHard baseline achieves close to  90.
> > >
> > > Oh, that's surprising. But I believe there should be a more "natural" (without using specific filtering mechanisms that might introduce bias), challenging math benchmark (e.g. AIME). Would you have time to explore it?

---

> > > > ### Author Response · Authors · 2024-12-04
> > > >
> > > > Thank you for taking the time to consider and reply to our initial responses! Please find our new responses to your individual questions below.
> > > >
> > > > > **just checking: do the current experiments evaluate "subtle differences" beyond correctness?**
> > > >
> > > > No, we currently do not explicitly evaluate the annotators’ ability to distinguish more subtle style differences. Building on existing datasets, the datasets used in our experiments focus on functional correctness. We consider good performance on functional correctness annotation as a pre-requisite to be a good LLM-as-a-Judge method, but would be excited for future work on datasets that also contain response pairs that only differ in style (but where both responses are functionally correct).
> > > >
> > > > > **I'm focusing on the degradation in iid setting (Figure 5) instead of ood.**
> > > >
> > > > Thank you for clarifying! Yes, we agree that challenges do also remain in the Math IID setting, in particular relative to the ArenaHard baseline shown in Figure 5. Effective tool-use for math tasks remains a challenge. To help inform further research, we conduct additional experiments to improve our understanding of our framework's existing math capabilities and their limitations, discussed below.
> > > >
> > > > > **Oh, that's surprising. But I believe there should be a more "natural" (without using specific filtering mechanisms that might introduce bias), challenging math benchmark (e.g. AIME). Would you have time to explore it?**
> > > >
> > > > Yes, we were able to explore a more challenging math benchmark: the recent RewardMATH dataset [4] suggested by Reviewer pD8T. According to the authors, RewardMATH may be considered as more challenging than the original RewardBench math subset (Figure 10), which they suggest may be susceptible to reward hacking due to the consistently lower number of solution steps in the correct vs incorrect solutions. RewardMATH uses a more diverse set of responses (coming from different models) than the original RewardBench math tasks.
> > > >
> > > > **Setup.** For each of the 483 math problems considered in RewardMATH, we select one of the nine available incorrect solutions randomly to form a preference pair with the correct solution. Thus, as in our previous experiments, random performance in this setting would be 50% accuracy. Baseline results are averaged over 5 seeds, agent results over a single seed (due to time constraints). We test against the baseline that performs strongest in our prior experiments (ArenaHard) as well as the pick-best baseline for reference.
> > > >
> > > > | Method                                             	| Accuracy  |
> > > > | ------------------------------------------------------ | --------- |
> > > > | Pick-best baseline (GPT-4o)                        	| 75.41 	|
> > > > | Agent (GPT-4o, tools: fact+code+math, base: pick-best) | **92.75** |
> > > > | ArenaHard baseline (GPT-4o)                        	| 87.91 	|
> > > > | Agent (GPT-4o, tools: fact+code+math, base: ArenaHard) | 92.55 	|
> > > > *Table A: Results on RewardMath*
> > > >
> > > > **Results:** Shown in Table A above, our agents are able to consistently outperform the baseline methods on this new math benchmark. Indeed we observe more of a gap than on the RewardBench math or GSM8k hard benchmarks, indicating that our method's capabilities are perhaps better-suited for the harder tasks of RewardMATH.
> > > >
> > > > *Why not AIME?* We note that our existing math experiments actually already include AIME questions, although retracing this origin requires multiple reasoning steps: we have an experiment with the math (Math-PRM) subset of RewardBench (Figure 10). This Math-PRM dataset is based on PRM800k [1] which is based on MATH [2,3] which (amongst other sources) contains AIME questions.
> > > >
> > > > We would of course add the discussion above as well as all experimental results to a camera-ready version.
> > > >
> > > >
> > > > - [1] [https://github.com/openai/prm800k](https://github.com/openai/prm800k)
> > > > - [2] [https://github.com/hendrycks/math](https://github.com/hendrycks/math)
> > > > - [3] https://arxiv.org/abs/2103.03874
> > > > - [4] https://arxiv.org/abs/2410.01729
> > > >
> > > > ---
> > > >
> > > > Finally, regarding your prior question on adding open-source models: due to time-constraints and the complexity of the other experiments, we were unable to add such models yet, but would aim to do so in a potential camera-ready version to aid reproducibility.

---

> > > > > ### Author Response · Authors · 2024-12-04
> > > > > **Update: New Themis Experimental Results**
> > > > >
> > > > > In addition to the responses above, we were able to add experimental baseline Themis results for a number of our experiments to put our framework into context.
> > > > >
> > > > > **Setup.** We ran the Themis model [1] on the LongFact (Figure 4), GSM8k (Figure 5) and the RewardBench (RB) OOD datasets (Figure 7). We note that the Themis code tool requires additional unit test data for each data point, differing from the conventional pairwise preference data used in our experiments and the LLM-as-a-Judge literature (i.e., response 1 + response 2 + preference label (+ prompt)). Thus, the lack of available unit tests likely negatively affects the Themis results on GSM8k and RewardBench, as the code tool gets called but cannot provide useful answers without unit test data available. We mark Themis results that may be affected by this issue with a star symbol (\*). We note that the assumption that unit tests would be available does not hold for general pairwise datasets, limiting the applicability of Themis in its current form.
> > > > >
> > > > >
> > > > > | Method                                                    	| LongFact (Figure 4) | GSM8k (Figure 5) | RB OOD (Figure 7) |
> > > > > | ------------------------------------------------------------- | ------------------- | ---------------- | ----------------- |
> > > > > | Themis-7B **(new)**                                       	| 48.00           	| 47.41\*      	| 49.41\*       	|
> > > > > | Simplest pick-best baseline (GPT-3.5-Turbo)               	| 51.00           	| 42.59        	| 64.82         	|
> > > > > | Agent (GPT-3.5-Turbo, tools: fact+code+math, base: pick-best) | 61.80           	| 54.83        	| 65.64         	|
> > > > > | Simplest pick-best baseline (GPT-4o)                      	| 63.00           	| 43.62        	| 84.92         	|
> > > > > | Agent (GPT-4o, tools: fact+code+math, base: pick-best)    	| **80.60**           	| 55.34        	| 83.23         	|
> > > > > | ArenaHard baseline (GPT-4o)                               	| 78.20           	| **63.28**    	| **86.75**     	|
> > > > > | Agent (GPT-4o, tools: fact+code+math, base: ArenaHard)    	| 80.40       	| 55.34        	| 84.99         	|
> > > > > *Table B: Results of Themis on our data*
> > > > >
> > > > > **Results.** In the results shown in Table B, we observe that Themis performs substantially worse than all our agents and most of our baselines. Themis performs close to a random annotator across the datasets tested (50%). Given the method's poor performance on our datasets, we actively tried to find potential flaws with our Themis setup but were only able to find the issue with the unit test data described above. As always, despite our best efforts, we cannot completely exclude the possibility that another issue was affecting Themis' performance.
> > > > >
> > > > > Overall, these results are in line with our previous results that indicate a notable performance difference between generally less capable models (GPT-3.5-Turbo) and more capable models (GPT-4o) in terms of both their baseline and tool-using annotation capability. As Themis is based on Llama-3 8B, it is significantly less capable than GPT4o (and somewhat less capable than GPT-3.5-Turbo), as measured by public benchmarks. We consider these observations supportive of a more flexible framework like the one we present here, that is able to leverage the latest frontier models whether open-source or accessed via API.
> > > > >
> > > > > **Additional experiment details.** We also tried to run Themis on the APPS dataset but ran into memory errors likely due to the larger context size required. We were unable to debug this issue in time for this rebuttal. Thus, we omit the APPS dataset here but note that we would not expect very strong performance based on the performance on the other datasets and the missing unit tests. Further, we additionally ran Themis on the RewardBench code datasets (Figure 9 in Appendix A) and obtained a performance of 49.49%, close to random (50%).
> > > > >
> > > > > - [1] https://huggingface.co/baidu/Themis-7b
> > > > >
> > > > > ---
> > > > >
> > > > > All experimental results above would of course be included in a camera-ready version. We hope that you will be able to consider these new experimental results in your final review.

---

### Official Review · Reviewer_pD8T · 2024-11-04

**Soundness:** 3
**Presentation:** 3
**Contribution:** 2
**Rating:** 5
**Confidence:** 5

**Summary:**

The paper proposes the use of external tools to create higher quality AI annotation systems and introduces a tool using AI annotator that use web-search and code execution to improve annotations. After establishing that existing annotation benchmarks are saturated, they introduce 3 new annotations datasets for fact checking, coding and mathematics. They demonstrate the efficacy of their tool-based AI annotator, by showing better performance on the 3 new datasets over SoTA AI annotators, while performing roughly on par with existing annotation benchmarks.

**Strengths:**

1. While the use of tools in AI-based applications is fairly commonplace now, there use for annotation system is an interesting and novel idea and the paper demonstrates fairly well that it works for a few domains at least.
2. The paper is well-written and presents fair experimental backing to its claims.
3. The paper introduced 3 novel datasets for evaluating domain specific annotation capabilities of Language models

**Weaknesses:**

1. While the use of toolings for AI annotators is interesting, in the current iteration of the work, it is not very clear if it will scale with more custom toolings. In the agent evaluator discussed in the paper, eventhough it defaults to existing annotations for the no-tool use cases, the system shows a degradation in performance for RewardBench, the only OOD dataset evaluated. This makes me concerned about the generalizability of the system.
2. Two of the proposed benchmarks don't have baseline human annotation scores, making it hard to quantify the degree of hardness of the datasets.
3. It is not very clear what are the advantages of using the agentic architecture compared to something like tool-calling API by OpenAI.

**Questions:**

1. For the generalizability issue, one suggestion would be to experiment with more recent and challenging open domain datasets like RMbench (https://arxiv.org/pdf/2410.16184) and external domain specific datasets like RMMath ( https://arxiv.org/pdf/2410.01729) to verify if the RewardBench results are an exception or a fundamental limitation of the technique- helping verify the robustness of the system.
2. Can the authors compare function calling API based tool-calling system with the existing implementation?

---

> ### Author Response · Authors · 2024-11-25
>
> Thank you for your detailed review! Please find below our responses to each of the weaknesses (W) and questions (Q) raised. All references refer to the updated manuscript, where we highlight relevant changes in blue.
>
> > **W1 (1): While the use of toolings for AI annotators is interesting, in the current iteration of the work, it is not very clear if it will scale with more custom toolings.**
>
>  We intentionally designed our framework such that it is able to scale with additional and more custom toolings. In particular, the *initial domain assessment* step of our method makes it possible to restrict the domain of any new tool to a well-defined task domain where the developer is confident that the tool improves performance. Thus, to add a new tool, we recommend two steps: (1) running the annotator with the tool on a broad range of annotation tasks where the tool *may* help; and (2) writing questions for the *initial domain assessment* that reliably classify tasks where the tool is empirically shown to help. This way, the initial domain assessment enables adding custom tools whilst limiting negative impact on out-of-domain tasks. We have added a clarification to Section 3.1 highlighting this motivation.
>
> > **W1 (2): In the agent evaluator discussed in the paper, eventhough it defaults to existing annotations for the no-tool use cases, the system shows a degradation in performance for RewardBench, the only OOD dataset evaluated. This makes me concerned about the generalizability of the system.**
>
> Thank you for raising this concern! We would like to clarify two aspects:
>
> 1. *RewardBench is more than one dataset.* Whilst accessible via a single download, RewardBench is an aggregate dataset combining a large set of existing datasets. In particular, the OOD subset we used consists of 16 separate sub-datasets covering a diverse set of domains [1].
>
> 2. *RewardBench OOD is a challenging task.* In the construction of this dataset, we explicitly omit all RewardBench subsets from categories where we would expect our method to potentially improve performance (math and coding tasks). We have clarified the construction in Section 4.4. As such, the dataset presents a very challenging task, where our tools and framework most likely cannot improve performance. Further, many of the remaining subsets of RewardBench are fairly saturated. On many datasets included in RewardBench our baseline annotators achieve $90\%+$ agreement. On such saturated datasets, the primary goal is preserve performance as well as possible --- rather than improve performance, as there are, relatively speaking, few datapoints to improve performance on. Overall, we note that this OOD test is likely a more challenging setting than in the real-world, where we would expect more of a mix of tasks that are similar to our own datasets, RewardBench datasets and other tasks.
>
> That said, we agree the OOD performance is an important area for future improvements and we would be excited to see future work in this area!
>
> - [1] Full list of included datasets: AlpacaEval Easy, AlpacaEval Length, AlpacaEval Hard, MT Bench Medium, MT Bench Hard, LLMBar Natural, LLMBar Adver. Neighbor, LLMBar Adver. GPTInst, LLMBar Adver. GPTOut, LLMBar Adver. Manual, Refusals Dangerous, Refusals Offensive, XSTest Should Refuse, XSTest Should Respond, Do Not Answer. See Table 1 of RewardBench paper (https://arxiv.org/abs/2403.13787)
>
> > **W2: Two of the proposed benchmarks don't have baseline human annotation scores, making it hard to quantify the degree of hardness of the datasets.**
>
> We had limited capacity for human annotation collection within this project. Given this limited capacity, we decided that long-form fact-checking would be the most interesting to benchmark against human annotators, but agree that future work providing human annotations across all datasets would be insightful.

---

> > ### Author Response · Authors · 2024-11-25
> >
> > > **W3: It is not very clear what are the advantages of using the agentic architecture compared to something like tool-calling API by OpenAI.**
> >
> > We agree that further comparison would be useful and, following your suggestion, **conducted an additional set of experiments comparing our framework to the standard tool-calling API by OpenAI**. We discuss the results in detail in the newly added Appendix B of our updated manuscript, but also summarize the results below.
> >
> > We adapt the simple *pick-best* and best-performing *ArenaHard* baselines to use a tool-enabled GPT-4o. We give GPT-4o access to two tools: code interpreter and web-search. In this setup, GPT-4o now has equivalent access to external validation as our framework, but lacks the additional scaffolding around when and how to use the tools our framework provides. We test this setup across four datasets considered in our paper: LongFact, GSM8k, APPS and RewardBench Out-of-Domain.
> >
> > We make the following observations: (1) using the tool-calling API without additional scaffolding does not notably improve performance across any of the tested datasets and LLM-as-a-Judge configurations, often it performs worse that the baseline without tool access; and (2) adding tools reduces the output reliability of GPT-4o-based ArenaHard baseline.
> >
> > Overall, the results appear to support our framework's use of more hand-crafted scaffolding around tool-use. In practice, we found that deciding when and how to effectively use a tool appear to be a fairly challenging tasks for current state-of-the-art LLMs like GPT-4o. Our framework partially removes the requirement to make all these decisions from scratch. For example, by using simpler questions about the task domain to then determine tool activation, rather than letting the model do this end-to-end.
> >
> > > **Q2: Can the authors compare function calling API based tool-calling system with the existing implementation?**
> >
> > Yes, absolutely. See response to W3.
> >
> >
> > > **Other questions**
> >
> > We are currently still working on responding to your remaining question (Q1).

---

> > > ### Author Response · Authors · 2024-12-02
> > >
> > > Thank you again for taking the time to review our work! With the discussion period ending soon (less than a day for reviewer comments), we wanted to briefly follow up to ask if there are any remaining concerns or questions regarding our response above?

---

> > > > ### Author Response · Authors · 2024-12-04
> > > > **Update: New Experimental Results**
> > > >
> > > > Thank you again for your detailed review! We are glad to be able to share an update regarding your Question 1 as promised:
> > > >
> > > > > **Q1: For the generalizability issue, one suggestion would be to experiment with more recent and challenging open domain datasets like RMbench ([https://arxiv.org/pdf/2410.16184](https://arxiv.org/pdf/2410.16184)) and external domain specific datasets like RMMath ( [https://arxiv.org/pdf/2410.01729](https://arxiv.org/pdf/2410.01729)) to verify if the RewardBench results are an exception or a fundamental limitation of the technique- helping verify the robustness of the system.**
> > > >
> > > > We were able to conduct additional experiments to evaluate our method on the suggested RMMath/RewardMATH dataset [1].
> > > >
> > > > **Setup.** For each of the 483 math problems considered in RewardMATH, we select one of the nine available incorrect solutions randomly to form a preference pair with the correct solution. Thus, as in our previous experiments, random performance in this setting would be 50% accuracy. According to the authors, RewardMATH may be considered as more challenging than the original RewardBench math subset (Figure 10), which they suggest may be susceptible to reward hacking due to the consistently lower number of solution steps in the correct vs incorrect solutions. Baseline results are averaged over 5 seeds, agent results over a single seed (due to time constraints). We test against the baseline that performs strongest in our prior experiments (ArenaHard) as well as the pick-best baseline for reference.
> > > >
> > > > | Method                                             	| Accuracy  |
> > > > | ------------------------------------------------------ | --------- |
> > > > | Pick-best baseline (GPT-4o)                        	| 75.41 	|
> > > > | Agent (GPT-4o, tools: fact+code+math, base: pick-best) | **92.75** |
> > > > | ArenaHard baseline (GPT-4o)                        	| 87.91 	|
> > > > | Agent (GPT-4o, tools: fact+code+math, base: ArenaHard) | 92.55 	|
> > > > *Table A: Results on RewardMath*
> > > >
> > > > **Results:** Shown in Table A above, our agents are able to consistently outperform the baseline methods on this new math benchmark. Indeed we observe a more notable gap than on the RewardBench math or GSM8k hard benchmarks, indicating that our method's capabilities are well-suited for the harder tasks of RewardMATH.  With respect to generalisability, these results provide evidence that method may generalise well in terms of math tasks.
> > > >
> > > > - [1] [https://arxiv.org/pdf/2410.01729](https://arxiv.org/pdf/2410.01729)
> > > >
> > > > ---
> > > >
> > > > All experimental results above would of course be included in a camera-ready version. We hope that you will be able to consider these new experimental results in your final assessment of our work.

---

### Comment · Area_Chair_29ZZ · 2024-11-27

Dear Reviewers,

Thank you for your efforts reviewing this paper. Can you please check the authors' response and see if your concerns have been addressed? Please acknowledge you have read their responses. Thank you!

---

### Author Response · Authors · 2024-12-04
**Discussion Summary**

We again thank all reviewers for their helpful and constructive feedback!

With the discussion period coming to an end, we wanted to provide a short summary of the main concerns raised in the reviews and our responses to address them:

---

**1. Additional math benchmarks** (Z3LQ, pD8T)

Z3LQ and pD8T recommended evaluating our framework on a more challenging math benchmark. In response, we conducted additional experiments on the recent RewardMATH benchmark, suggested by pD8T. We observe our framework outperforming all tested baselines on this benchmark.

---

**2. Comparison to standard tool-calling models** (pD8T, Qxh9)

pD8T raised concerns how our framework improves over standard tool-calling models, such as the OpenAI tool-calling API. Relatedly, Qxh9 raised questions regarding the impact of individual agentic components. In response, we tested additional baselines across the main experiments: GPT-4o-based annotators with standard OpenAI API tools. Overall, the tested tool-using baselines perform worse than our tool-using framework.

---

**3. Comparison to related tool-augmented AI feedback work** (Z3LQ)

Z3LQ brought two related papers to our attention that also use tool-augmented models to provide feedback. In response, we included an extensive discussion of the similarities and differences to our framework both in the rebuttal and our related work section. Further, we conducted additional experiments to compare our framework to the directly related method (Themis), finding our approach outperforming the prior work.

---

**4. Potential issues with GSM8k data** (jhb8)

jhb8 recommended closely checking the correctness of our GSM8k hard answers, highlighting known issues with the underlying dataset. In response, we conducted a thorough manual check of our entire GSM8k hard dataset and only found two incorrect datapoints (less than 2% of the data) — too few to substantially affect our results.

---

Beyond the major concerns outlined above, there were also a number of smaller issues and comments that we addressed. To keep this summary concise, we refer back to the individual rebuttals for full details. Further, any experimental results we shared after the manuscript submission closed would of course be included and discussed in a camera-ready version.

To conclude, we again thank all reviewers for their time and help in improving the paper!

---

### Meta-Review · Area_Chair_29ZZ · 2024-12-22

**Metareview:**

Summary:

This paper explores augmenting standard AI annotator systems with additional tools to improve performance on three challenging response domains: long-form factual, math and code tasks. The proposed system uses web-search and code execution to be grounded on external validation. The paper conducted experiments using three targeted response domains as well as general annotation tasks, using RewardBench data (including AlpacaEval and LLMBar), as well as three new datasets for areas where existing datasets are saturated. The results indicate that external tools can indeed improve AI annotator performance in many, but not all, cases, and highlight the high variability of AI annotator performance w.r.t. simple parameters (e.g., prompt) and the need for improved (non-saturated) annotator benchmarks.

Strengths:

1. Reviewers generally agree that while the use of tools in AI-based applications is fairly commonplace now, their use for annotation systems is an interesting idea and the paper demonstrates fairly well that it works for a few domains at least.

2. The paper’s effort to extend subsets of RewardBench to create more challenging test sets is appreciated.

3. the paper introduced three new datasets for evaluating domain specific annotation capabilities.

Weaknesses:

1. I think the author response to the question (raised by pD8T) regarding RewardBench OOD and the generalizability of the proposed system is still confusing. According to the response, “this OOD test is likely a more challenging setting than in the real-world”, this seems to mean this OOD test is less realistic, and in addition, “the dataset presents a very challenging task, where our tools and framework most likely cannot improve performance.” Then it is not very clear why the paper chooses to include this test set to demonstrate OOD generalization.

2. Reviewer Qxh9 and Z3LQ share some concern on the novelty of the proposed work: there are some highly related work that is missed in the original submission. From the author response, it is still not very convincing why some of these methods (especially CRITIC) cannot be compared with the proposed framework.

**Additional Comments On Reviewer Discussion:**

There are many comments from the reviewers that were addressed during the rebuttal period. For example,

1. Additional math benchmarks. Reviewer Z3LQ and pD8T recommended evaluating the proposed framework on a more challenging math benchmark. In response, the authors conducted additional experiments on the recent RewardMATH benchmark, suggested by pD8T.

2. Reviewer jhb8 raised some minor questions that have been addressed during the discussion period.

---

### Decision · Program_Chairs · 2025-01-22

Reject